# RAGRouter: Learning to Route Queries to Multiple Retrieval-Augmented Language Models

Jiarui Zhang[1]   Xiangyu Liu[2]   Yong Hu[2]   Chaoyue Niu[1]*   Fan Wu[1]   Guihai Chen[1]

[1]Shanghai Jiao Tong University   [2]WeChat, Tencent Inc

## Abstract

Retrieval-Augmented Generation (RAG) significantly improves the performance of Large Language Models (LLMs) on knowledge-intensive tasks. However, varying response quality across LLMs under RAG necessitates intelligent routing mechanisms, which select the most suitable model for each query from multiple retrieval-augmented LLMs via a dedicated router model. We observe that external documents dynamically affect LLMs' ability to answer queries, while existing routing methods, which rely on static parametric knowledge representations, exhibit suboptimal performance in RAG scenarios. To address this, we formally define the new retrieval-augmented LLM routing problem, incorporating the influence of retrieved documents into the routing framework. We propose RAGRouter, a RAG-aware routing design, which leverages document embeddings and RAG capability embeddings with contrastive learning to capture knowledge representation shifts and enable informed routing decisions. Extensive experiments on diverse knowledge-intensive tasks and retrieval settings, covering open and closed-source LLMs, show that RAGRouter outperforms the best individual LLM and existing routing methods. With an extended score-threshold-based mechanism, it also achieves strong performance-efficiency trade-offs under low-latency constraints. [2]

## 1  Introduction

The rapid advancement of large language models (LLMs) has led to an increasingly diverse model landscape, with significant heterogeneity in parametric knowledge stemming from variations in training data, architectures, and learning objectives [45, 35, 36, 3, 4, 57, 53]. However, LLMs remain limited by outdated knowledge, hallucinations, and insufficient domain coverage [17, 22, 29]. Retrieval-Augmented Generation (RAG) [27, 15] addresses these issues by injecting external knowledge at inference time, effectively reducing hallucinations and improving performance on knowledge-intensive tasks [5, 13, 47, 20, 41].

While RAG enhances LLM performance by incorporating external knowledge, different LLMs exhibit substantial variation in their ability to utilize retrieved content. Prior studies [6, 11] show that, given identical documents, LLMs differ in information extraction, integration, and robustness to noise—reflecting inherent heterogeneity in RAG capabilities stemming from differences in architecture, training data, and optimization. Such diversity suggests that combining multiple models can yield complementary strengths, enabling performance that surpasses any single model. LLM routing, which pre-selects the most suitable model for each query from a pool of LLMs without invoking them all, offers an efficient and effective fusion strategy [9]. Under dual heterogeneity in parametric knowledge and RAG capability, intelligent query routing presents a promising direction for leveraging RAG to achieve superior performance, motivating the central question: *How can we effectively route queries to the most capable LLM under the RAG paradigm?*

---

*Chaoyue Niu is the corresponding author (rvince@sjtu.edu.cn).

[2]The code and data are available at `https://github.com/Oww099/RAGRouter`.

39th Conference on Neural Information Processing Systems (NeurIPS 2025).

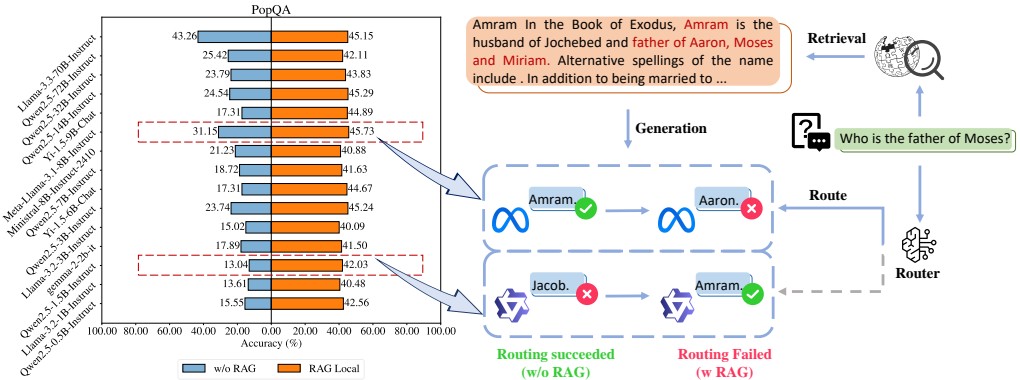

Figure 1: **Left**: Accuracy of various LLMs on the PopQA task before and after RAG. **Right**: An example query where retrieved documents improve Qwen's response (unanswerable → answerable) but impair Llama's (answerable → unanswerable), illustrating how existing routing methods fail under RAG due to their inability to capture such dynamic shifts.

Current multi-model routing methods primarily match queries to LLMs based on their inherent parametric knowledge [40, 19, 31, 43, 38, 10, 7, 33, 58, 12]. Several approaches [7, 33, 58, 12] construct compact vector representations of LLMs to enable efficient query-model compatibility estimation. These methods all assume static knowledge representations for non-RAG scenarios.

However, these approaches face critical limitations in RAG settings, as they fail to account for the dynamic impact of knowledge injection. As illustrated in Figure 1, RAG dramatically shifts the distribution of response quality across input queries—external documents can reverse a model's ability to answer a question, rendering routing strategies designed for non-RAG scenarios obsolete. The core issue lies in the **Static Knowledge Assumption**: existing approaches assume fixed LLM knowledge, ignoring how retrieved content dynamically reshapes their capabilities. In practice, RAG response quality depends on the interplay between a model's internal knowledge and external information. This leads to shortcomings: **Missing Doc Interaction**—existing methods focus on queries and model embeddings, overlooking document features and their interaction with models; and **Ignoring RAG Capability**—prior work captures only static knowledge differences, neglecting LLMs' differing ability to leverage documents. These gaps highlight shortcomings of current LLM routing strategies—they fail to adapt to the dynamic relationship between LLM and external knowledge.

To tackle this issue, we propose **RAGRouter**, a contrastive learning-based routing framework that explicitly models knowledge shifts in RAG scenarios. RAGRouter is designed to route queries across LLMs by modeling key factors that affect post-retrieval performance. At the **architecture level**, RAGRouter incorporates a document encoder and a cross encoder to capture document semantics and query interactions, thereby addressing missing document interaction, and assigns each LLM a RAG capability embedding—a learnable vector representing its proficiency in utilizing retrieved content—to mitigate ignoring RAG capability. However, directly optimizing such a router is challenging due to inherent variations introduced by retrieval. To address this, at the **optimization level**, we employ a contrastive learning objective, where positive and negative samples—i.e., representations of LLMs that correctly or incorrectly respond to a query—are drawn from both *Cross-Setting* (between non-RAG and RAG settings) and *Intra-Setting* (within each setting). Taking the query representation as an anchor, the objective encourages alignment between answerable model-query pairs while pushing apart unanswerable ones. This allows RAGRouter to effectively model retrieval-induced behavior shifts, moving beyond the static knowledge assumption.

We evaluate RAGRouter on a suite of knowledge-intensive tasks [32, 34, 25, 2, 23] and retrieval settings. Experimental results show that RAGRouter surpasses the performance of the best individual LLM, highlighting its ability to leverage the complementary strengths of multiple models in retrieval-augmented scenarios. Furthermore, RAGRouter substantially outperforms existing non-RAG-aware routing methods, validating the effectiveness of modeling retrieval-induced knowledge shifts.

Our main contributions are summarized as follows: (i) To the best of our knowledge, this is the first work exploring LLM routing in the RAG setting; (ii) We propose RAGRouter, a contrastive learning-based routing mechanism that is aware of knowledge shifts, incorporating RAG capability and document-aware representations to effectively address the failure modes of existing routing strategies in RAG; (iii) We validate the effectiveness of our method on five knowledge-intensive tasks

under local and online retrieval settings, using open-source LLMs such as the Qwen and LLaMA series scales from 0.5B to 72B and closed-source LLMs including GPT-4o, Qwen2.5-Max, and DeepSeek-R1. Results show that RAGRouter outperforms the best individual LLM and existing non-RAG-aware routing methods by 1.67%–9.33%; (iv) We apply an extended score-threshold-based mechanism to RAGRouter, and results show that its accuracy–latency curve generally lies above those of all baselines, indicating superior performance-efficiency trade-offs under low-latency constraints.

## 2 Related Work

**Retrieval-Augmented Generation.** RAG enhances language models by integrating retrieved information from external databases [27, 15]. It typically follows a round of retrieval and sequential generation pipelines, where documents are retrieved based on the input query and concatenated with it for generation. Prior work has improved RAG by optimizing retrieval components [56, 51, 42] or enhancing the generator's ability to utilize retrieved content [21, 49, 48]. Recent studies highlight heterogeneous LM capabilities in processing external information, both in utilizing retrieved content [28, 6] and tolerating retrieval noise [11, 39]. These heterogeneous capabilities reveal optimization opportunities for ensemble approaches that strategically leverage multiple LLMs within RAG scenarios. In this work, we study the routing problem under the RAG setting.

**LLM Routing.** Existing LLM routing approaches [7, 10, 12, 33, 58, 31] primarily focus on non-RAG settings, where routing relies solely on the input query and each model's parametric knowledge, without incorporating external retrieved documents. For example, RouterDC [7] uses dual contrastive learning to model query-model compatibility, while EmbedLLM [58] and RouteLLM [33] apply matrix factorization to learn compact model embeddings for scalable routing. GraphRouter [12] constructs a heterogeneous graph with nodes for tasks, queries, and LLMs, and encodes their interactions as edges to capture contextual alignment between query needs and model capabilities. However, in RAG scenarios, retrieved documents induce dynamic shifts in model knowledge, which existing methods overlook. In contrast, our proposed RAGRouter models both the documents and LLMs' RAG capabilities, enabling more effective routing under retrieval-augmented settings.

## 3 Problem Formulation

RAG enhances LLMs by integrating external knowledge through a two-stage process: given a query $q$, the retriever $\text{Ret}(\mathcal{D}, q)$ selects relevant documents $d$ from an external corpus $\mathcal{D}$, and the model $M(q, d)$ generates a response $y$ based on both the query $q$ and the documents $d$, i.e., $y = M(q, \text{Ret}(\mathcal{D}, q))$.

We formulate a LLM routing problem under RAG setting. Let $\mathcal{M} = \{M_1, \ldots, M_N\}$ be a set of candidate LLMs. A routing policy $R : \mathcal{Q} \times \mathcal{D} \to \{1, \ldots, N\}$ selects the most suitable model $M_{R(q,d)}$ for each input pair $(q, d)$. To evaluate response quality, we define an oracle scoring function $\sigma(M_i, q, d) \in \{0, 1\}$, where $\sigma(M_i, q, d) = 1$ if the response from $M_i$ given $q$ and $d$ matches the reference answer $y^*$. Importantly, using a fixed model can be suboptimal, as different LLMs excel on different query-document pairs. The objective is to maximize the expected routing performance:

$$\max_R \mathbb{E}_{q \sim \mathcal{Q}} \left[ \sigma(M_{R(q,d)}, q, d) \right] \tag{1}$$

Notably, when no external documents are available (i.e., $d = \emptyset$), the LLM routing problem under RAG setting naturally degenerates into the conventional LLM routing problem. In this setting, the routing policy simplifies to $R : \mathcal{Q} \to \{1, \ldots, N\}$, and the oracle scoring function becomes $\sigma(M_i, q)$, which assesses the response based solely on the query. The objective becomes:

$$\max_R \mathbb{E}_{q \sim \mathcal{Q}} \left[ \sigma(M_{R(q)}, q) \right] \tag{2}$$

Thus, the conventional routing problem can be seen as a special case of LLM routing under the RAG setting, corresponding to the boundary condition where $d = \emptyset$.

## 4 RAGRouter

### 4.1 Routing Model Architecture Design

We establish a conceptual framework by constructing an intuitive explanation of knowledge representation and LLM-query matching under RAG and non-RAG settings. In non-RAG settings

[7, 58], each LLM is typically associated with a compact knowledge representation vector $v_k \in \mathbb{R}^{\dim}$, which implicitly reflects its parametric knowledge; and meanwhile, a query is encoded as $v_q \in \mathbb{R}^{\dim}$, representing the knowledge needed to answer it. A proxy metric, like similarity $\text{sim}(v_q, v_k)$ is then used to gauge the LLM's ability to respond, guiding non-RAG routing process.

However, in RAG settings, the LLM is augmented with retrieved documents that provide non-parametric knowledge. This additional information influences the model's response generation, rendering the original knowledge representation $v_k$ insufficient. The effective knowledge of the LLM shifts due to the integration of external information, resulting in a new representation:

$$v'_k = v_k + v_f \tag{3}$$

where $v_f$ is the fused knowledge representation derived from the documents. Consequently, in RAG scenarios, the similarity between the query and the updated knowledge representation $\text{sim}(v_q, v'_k)$ should serve as the new routing criterion, as it more accurately reflects the model's ability to respond.

RAGRouter is designed with this insight in mind and explicitly models the fused knowledge $v_f$ to dynamically update the LLM's knowledge representation. We identify three core factors that influence $v_f$: (1) the non-parametric knowledge provided by the documents; (2) the LLM's ability to process external information, including knowledge extraction and robustness to noise; and (3) the query's role in guiding knowledge retrieval. Based on these, RAGRouter consists of the following modules, with its architecture illustrated in Figure 2.

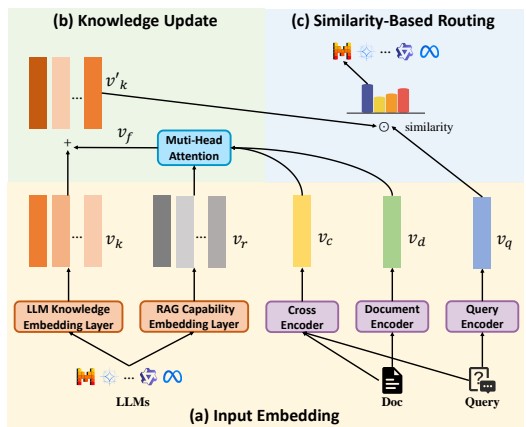

Figure 2: The inference pipeline of RAGRouter: (a) Encode query, document, cross interaction, LLM knowledge, and RAG capability; (b) Fuse RAG capability, document, and cross embeddings to update knowledge representation; (c) Route based on similarity with the query embedding.

**Representing Parametric Knowledge.** To obtain the original parametric knowledge representation $v_k$, we introduce the LLM Knowledge Embedding Layer $\phi_K$, which takes the LLM ID $M$ and outputs $v_k = \phi_K(M)$, capturing inter-model variability in parametric knowledge. For query representation, we employ a Query Encoder $\phi_Q$, which encodes the query $q$ as $v_q = \phi_Q(q)$.

**Representing RAG-Aware Factors.** To compute the fused knowledge $v_f$, RAGRouter integrates signals from three perspectives. First, **the non-parametric knowledge provided by the documents** is captured by the **Document Encoder** $\phi_D$, which encodes a document $d$ into $v_d = \phi_D(d)$. In practice, the document and query encoders share parameters to ensure consistency in the embedding space. Second, **the LLM's ability to process external information** is captured by the **RAG Capability Embedding Layer** $\phi_R$, which maps each candidate model $M$ to an embedding $v_r = \phi_R(M)$, representing its intrinsic capacity to utilize retrieved evidence. Third, **the query's role in guiding knowledge retrieval** is represented by the **Cross Encoder** $\phi_C$, which processes the query-document pair $(d, q)$ to produce an interaction representation $v_c = \phi_C(d, q)$.

**Representation Update for Similarity-Based Routing.** The fused knowledge representation $v_f$ is then derived via a multi-head attention mechanism that integrates these signals:

$$v_f = \text{Attention}(v_r, v_d, v_c) \tag{4}$$

With the fused knowledge computed, the RAG-aware knowledge representation of the model becomes $v'_k = v_k + v_f$. The final routing decision is based on the similarity between the query and the updated knowledge representations of candidate models:

$$R(q, d) = \arg \max_{i \in \{1, \dots, N\}} \{\text{sim}(v_q, v'_{k_i})\} \tag{5}$$

This formulation allows the routing policy to explicitly account for knowledge shifts introduced by document retrieval, thus maintaining accurate assessment of each LLM's ability in RAG settings.

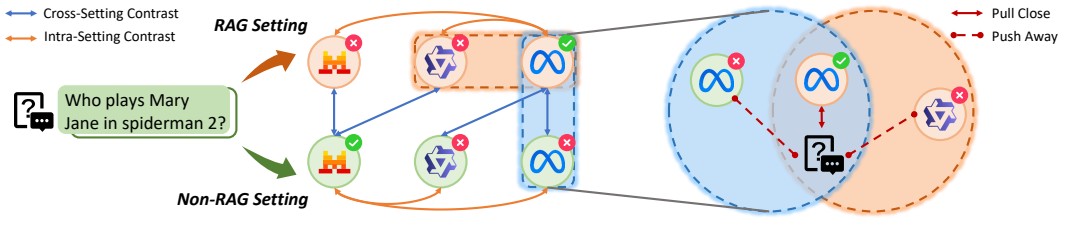

(a) Construct Positive and Negative Samples for One Query    (b) Contrastive Learning

Figure 3: (a) CSC constructs positive and negative samples under different settings based on response quality (e.g., Llama w/ RAG (✓) vs. Llama w/o RAG (✗)), while ISC constructs them under the same setting (e.g., Llama w/ RAG (✓) vs. Qwen w/ RAG (✗)); (b) By combining CSC and ISC, contrastive learning pulls positive samples closer to the query representation and pushes negative ones away.

## 4.2 Optimization

In RAG settings, the incorporation of retrieved documents often leads to significant changes in LLM answerability—some LLMs become able to answer queries they previously could not, while others fail after retrieval. These shifts in answerability, effectively label transitions, reflect corresponding changes in the model's knowledge representation. Such transitions naturally yield structured positive and negative pairs across different knowledge states. This setting aligns well with the principles of contrastive learning [8, 18], which is particularly well-suited for capturing and optimizing the knowledge representation shifts induced by external knowledge injection in RAGRouter.

To this end, we design the **Cross-Setting Contrast (CSC)** mechanism to model representation differences between the non-RAG and RAG settings, and introduce the **Intra-Setting Contrast (ISC)** mechanism to model representation differences within the same setting. As shown in Figure 3, using the query representation $v_q$ as the anchor, CSC constructs positive and negative samples by selecting knowledge representations with different response qualities from the non-RAG and RAG settings (blue arrows). ISC, on the other hand, selects positive and negative samples from models with different response qualities within the same setting (orange arrows). This enables CSC to help RAGRouter distinguish between different knowledge transfer patterns induced by documents, while ISC enhances the model's discriminative ability across LLMs within the same setting.

Combining CSC and ISC, we construct a comprehensive set of positive and negative samples to train the RAGRouter. For a given query $q$, we define the positive and negative sets as follows:

$$\begin{cases} V_+ = \{v_{k_i} \mid \sigma(M_i, q) = 1\} \cup \{v'_{k_j} \mid \sigma(M_j, d, q) = 1\} \\ V_- = \{v_{k_i} \mid \sigma(M_i, q) = 0\} \cup \{v'_{k_j} \mid \sigma(M_j, d, q) = 0\} \end{cases} \quad (6)$$

The corresponding contrastive loss is defined as:

$$\mathcal{L}_{CT}(q) = \sum_{v_{k+} \in V_+} - \log \frac{\exp(\text{sim}(v_q, v_{k+})/\tau)}{\exp(\text{sim}(v_q, v_{k+})/\tau) + \sum_{v_{k-} \in V_-} \exp(\text{sim}(v_q, v_{k-})/\tau)} \quad (7)$$

where $\tau$ is a temperature hyperparameter. This loss encourages the query embedding $v_q$ to be closer to positive samples and further from negative ones, enabling the learning of representations that are sensitive to both knowledge shifts and model heterogeneity. When retrieved documents alter a model's response ability—e.g., from unanswerable to answerable—the mechanism captures these dynamic transitions, enhancing routing accuracy and knowledge adaptability in RAGRouter.

To further enhance LLM discrimination, we introduce a binary classification loss. For the original model $M$, define $s_{M,q} = \text{Sigmoid}(\text{sim}(v_k, v_q))$; for the RAG-enhanced model $M'$, define $s_{M',q} = \text{Sigmoid}(\text{sim}(v'_k, v_q))$. Let $y_{M,q} = \sigma(M, q)$ and $y_{M',q} = \sigma(M, d, q)$ be the ground-truth labels. The classification loss is:

$$\mathcal{L}_{CLS}(q) = - \sum_{M \in \mathcal{M} \cup \mathcal{M}'} [y_{M,q} \log s_{M,q} + (1 - y_{M,q}) \log(1 - s_{M,q})] \quad (8)$$

The total loss is the weighted sum of the contrastive loss and classification loss, with $\lambda > 0$ as a balancing hyperparameter:

$$\mathcal{L}(q) = \mathcal{L}_{CT}(q) + \lambda \mathcal{L}_{CLS}(q) \quad (9)$$

### 4.3 Latency-Aware Extended Design

While RAGRouter does not explicitly model LLM's latency, it outputs a relevance score for each candidate LLM given a query, which can be exploited to support flexible trade-offs between performance and efficiency. To this end, we introduce a score-threshold-based routing mechanism. Concretely, we pre-sort the $N$ available LLMs as $[M_1, M_2, \ldots, M_N]$ based on their prior efficiency profiles, such as smaller parameter sizes and lower latency—meaning that $M_1$ is the most efficient and $M_N$ the least.

Given a query, suppose $M_i$ receives the highest predicted score from RAGRouter (i.e., it is the performance-optimal model). Instead of routing directly to $M_i$, we traverse the list from $M_1$ to $M_i$ and select the first LLM $M_j$ whose score satisfies $s_{M_i,q} - s_{M_j,q} \leq \theta$, where $\theta$ is a user-defined score margin threshold. This mechanism sacrifices a small amount of accuracy for significantly improved efficiency, making RAGRouter adaptable to latency-constrained or resource-limited scenarios.

## 5 Experiments

### 5.1 Experimental Setup

**Datasets.** We select queries from five different knowledge-intensive tasks: (i) PopQA [32] is an open-domain question-answering benchmark covering diverse factual topics from broad knowledge domains; (ii) MedMCQA [34] is a multiple-choice benchmark focused on biomedical knowledge and clinical reasoning; (iii) Natural Questions (NQ) [25] is an open-domain benchmark based on real-world search queries requiring span-level answer retrieval from Wikipedia; (iv) WebQuestions (WebQ) [2] is a knowledge base-driven benchmark grounded in Freebase relations, designed to evaluate entity-centric factual reasoning; and (v) TriviaQA (TQA) [23] is an open-domain benchmark centered on factoid-style questions sourced from trivia enthusiasts and web documents. Following [42], we adopt Cover Exact Match as the evaluation metric for PopQA, NQ, WebQ, and TriviaQA. Further data processing details and dataset statistics are summarized in Appendix A.2.

**Candidate LLMs.** We selected 15 mainstream LLMs [52, 14, 44, 55, 1] with parameter size ranging from 0.5B to 72B. Comprehensive statistics on model scales and latency [3] are presented in Table 1, and implementation details are provided in Appendix A.1.

**Retrieval Settings.** Following [42], we adopt both local and online retrieval strategies for PopQA and MedMCQA to reflect realistic RAG scenarios. Local retrieval uses the 2018 English Wikipedia dump [24] with BGE-large-en-v1.5 [50] as the dense retriever. Online retrieval leverages the DuckDuckGo Web Search API [4] to access up-to-date external content. For NQ, WebQ, and TriviaQA, we follow [11] and construct retrieval contexts from Wikipedia passages augmented with synthetic noise (e.g., irrelevant distractors, counterfactual noise) to simulate imperfect retrieval. This setting enables evaluate the effectiveness of the routing model under noisy conditions.

Table 1: Statistics of different LLMs and their latency.

| LLM | Params (B) | Latency (ms) |
|---|---|---|
| Qwen2.5-0.5B-Instruct | 0.494 | 24.54 |
| Llama-3.2-1B-Instruct | 1.240 | 20.47 |
| Qwen2.5-1.5B-Instruct | 1.500 | 24.79 |
| gemma-2-2b-it | 2.614 | 31.80 |
| Llama-3.2-3B-Instruct | 3.213 | 81.82 |
| Qwen2.5-3B-Instruct | 3.000 | 24.39 |
| Yi-1.5-6B-Chat | 6.061 | 142.67 |
| Qwen2.5-7B-Instruct | 7.616 | 80.83 |
| Ministral-8B-Instruct-2410 | 8.020 | 26.13 |
| Meta-Llama-3.1-8B-Instruct | 8.030 | 177.37 |
| Yi-1.5-9B-Chat | 8.829 | 199.61 |
| Qwen2.5-14B-Instruct | 14.770 | 175.42 |
| Qwen2.5-32B-Instruct | 32.764 | 156.26 |
| Qwen2.5-72B-Instruct | 72.706 | 1610.00 |
| Llama-3.3-70B-Instruct | 70.554 | 1970.00 |

**Baselines.** We compare RAGRouter against a range of baselines. Existing routing methods were not RAG-aware and exploited only query-LLM compatibility, ignoring the impact of retrieval augmentation. This series of baselines include **Prompt LLM** [12], which employs GPT-4o for model selection via meta-prompts; **GraphRouter** [12], which models queries, tasks, and LLMs in a heterogeneous graph; **RouterDC** [7], which aligns query-LLM embeddings through contrastive learning; **KNN Router**, which [19] relies on historical performance of similar queries; and **Matrix Factorization (MF)** [33, 58], which reconstructs LLM correctness patterns via low-rank latent spaces. We also introduce some rule-based routing methods, including a **Single Fixed LLM** for all queries; **Oracle Single Best** that ideally selects the best-performing single LLM per dataset; **Random** LLM

---

[3]Latency refers to the average time taken by an LLM to complete a single query, including both inference time and potential network delays.

[4]https://duckduckgo.com

Table 2: Performance comparison of RAGRouter with rule-based and non-RAG-aware baselines across different knowledge-intensive tasks and retrieval settings. Testing accuracy (%) is reported. "Ret." indicates whether the method is retrieval-aware (✓) or not (✗). The best results are shown in **bold**, and the second-best are underlined.

| Method | Ret. | PopQA | | MedMCQA | | NQ | WQ | TQA | Avg |
|---|---|---|---|---|---|---|---|---|---|
| | | Local | Online | Local | Online | | | | |
| Qwen2.5-0.5B-Instruct | - | 45.19 | 51.11 | 25.93 | 34.81 | 27.08 | 38.75 | 39.17 | 37.43 |
| Llama-3.2-1B-Instruct | - | 39.26 | 46.67 | 17.78 | 36.67 | 25.42 | 31.67 | 43.33 | 34.40 |
| Qwen2.5-1.5B-Instruct | - | 44.44 | 48.89 | 30.00 | 42.59 | 30.00 | 35.42 | 46.67 | 39.72 |
| gemma-2-2b-it | - | 41.11 | 50.74 | 20.37 | 37.04 | 22.92 | 27.92 | 40.83 | 34.42 |
| Llama-3.2-3B-Instruct | - | 45.56 | 51.48 | 27.41 | 44.81 | 36.67 | 45.42 | 65.00 | 45.19 |
| Qwen2.5-3B-Instruct | - | 41.11 | 47.41 | 40.37 | 49.26 | 30.42 | 36.25 | 53.33 | 42.59 |
| Yi-1.5-6B-Chat | - | 46.67 | 51.48 | 31.11 | 40.00 | 35.83 | 43.33 | 56.67 | 43.58 |
| Qwen2.5-7B-Instruct | - | 42.96 | 48.15 | 35.93 | 43.33 | 29.58 | 35.42 | 55.00 | 41.48 |
| Ministral-8B-Instruct-2410 | - | 41.48 | 46.30 | 50.74 | 62.22 | 38.33 | 42.08 | 63.75 | 49.27 |
| Meta-Llama-3.1-8B-Instruct | - | 46.67 | 51.85 | 41.85 | 52.59 | 39.58 | 46.25 | 69.58 | 49.77 |
| Yi-1.5-9B-Chat | - | 46.67 | **52.59** | 50.74 | 57.78 | 38.33 | 47.08 | 58.75 | 50.28 |
| Qwen2.5-14B-Instruct | - | 46.30 | 50.00 | 57.04 | 64.07 | 42.92 | 47.08 | 72.50 | 54.27 |
| Qwen2.5-32B-Instruct | - | 45.93 | 48.52 | 43.33 | 49.63 | 44.58 | 50.42 | 80.42 | 51.83 |
| Qwen2.5-72B-Instruct | - | 44.81 | 47.78 | 67.04 | 70.00 | 40.00 | 48.75 | 79.17 | 56.79 |
| Llama-3.3-70B-Instruct | - | 46.30 | 50.37 | 68.89 | 70.37 | 51.67 | 50.42 | 87.92 | 60.85 |
| **Oracle Single Best** | - | 46.67 | **52.59** | 68.89 | 70.37 | 51.67 | 50.42 | 87.92 | 61.22 |
| Random | - | 44.30 | 49.56 | 40.57 | 50.35 | 35.56 | 41.75 | 60.81 | 46.13 |
| Weighted | - | 46.35 | 50.53 | 68.31 | 70.18 | 46.87 | 48.39 | 86.09 | 59.53 |
| Prompt LLM [12] | ✗ | 46.67 | 51.48 | 61.85 | 65.93 | 39.58 | 49.17 | 71.25 | 55.13 |
| GraphRouter [12] | ✗ | 47.41 | 51.48 | 68.89 | 70.37 | 51.67 | 50.42 | 87.92 | 61.17 |
| RouterDC [7] | ✗ | 44.81 | 50.37 | 67.04 | 68.89 | 40.00 | 48.33 | 77.50 | 56.71 |
| KNN Router [19] | ✗ | 46.67 | 52.22 | 68.15 | 71.48 | 52.08 | 46.25 | 86.25 | 60.44 |
| MF [33, 58] | ✗ | 46.30 | **52.59** | 68.89 | 71.48 | 49.17 | 50.42 | 82.92 | 60.25 |
| **RAGRouter (Ours)** | ✓ | **48.52** | **52.59** | **71.48** | **74.44** | **56.67** | **56.67** | **90.83** | **64.46** |
| Oracle | - | 54.44 | 57.41 | 91.85 | 90.37 | 69.17 | 77.92 | 96.25 | 76.77 |

assignment; **Weighted** routing to different LLMs according to RAGRouter's empirical probability distribution. To show the ideal upper bound of routing performance, we introduce the **Oracle** baseline by routing each query to its optimal LLM using ground-truth performance data.

**Implementation Details.** For the RAGRouter architecture, we use all-mpnet-base-v2 [37] as the encoder for both queries and documents, and ms-marco-MiniLM-L12-v2 [46] as the cross-encoder, resulting in a total parameter size of approximately 136M. Both the knowledge representation vector and the RAG capability vector are set to a dimensionality of 768. To mitigate overfitting, all but the last two transformer layers in the query/document encoder and the cross-encoder are frozen during training. The classification loss weight $\lambda$ is set to 2.0, and the contrastive learning temperature $\tau$ to 0.2. The router is optimized using AdamW [30] with a learning rate of 5e-5, batch size of 64, for 10 epochs. All experiments are conducted on a single NVIDIA RTX 4090D GPU.

## 5.2 Main Results

**Comparison with Single Non-Routed LLMs.** Table 2 presents the test accuracy across a variety of knowledge-intensive tasks and retrieval settings. RAGRouter consistently achieves the highest performance, with an average accuracy of 64.46%. It surpasses the best-performing single RAG-enabled LLM, LLaMA-3.3-70B-Instruct (60.85%), by +3.61%, demonstrating the effectiveness of routing in RAG. Notably, RAGRouter also outperforms the Oracle Single Best baseline (61.22%)—which selects the optimal single model for each dataset—by +3.24%, indicating that routing enables the integration of multiple RAG-enhanced LLMs to achieve performance beyond any individual model.

**Comparison with Non-RAG-Aware and Other Baselines.** RAGRouter significantly outperforms all non-retrieval-aware routing baselines, including GraphRouter (61.17%, +3.29%), MF (60.25%, +4.21%), KNN Router (60.44%, +4.02%), and RouterDC (56.71%, +7.75%). These results underscore the limitations of methods that do not explicitly model the interaction between LLMs and retrieved

knowledge. Without capturing retrieval-induced capability shifts, such approaches struggle to make effective routing decisions in RAG. RAGRouter also surpasses rule-based strategies such as Random (46.13%, +18.33%) and Weighted (59.53%, +4.93%). Together, these findings support our core claim: modeling RAG-specific capabilities and knowledge shift is essential for accurate LLM routing.

**Inference Cost of RAGRouter.** The inference characteristics are evaluated on a single NVIDIA RTX 4090D GPU. Peak GPU memory usage is 4147 MiB. With a batch size of 64, RAGRouter processes 270 instances in 3 seconds across 5 batches, yielding an average inference time of 0.011 seconds per instance. These results clearly show that RAGRouter is lightweight and efficient during inference.

## 5.3 Routing Performance Involving Closed-Source LLMs

To verify that RAGRouter can still deliver performance gains with strong closed-source LLMs, we augment the candidate LLMs set—comprising Qwen2.5-32B-Instruct, Qwen2.5-72B-Instruct, and Llama-3.3-70B-Instruct—with closed-source models Qwen2.5-Max [5], GPT-4o [6], and reasoning-based DeepSeek-R1 [7].

As shown in Table 3, RAGRouter outperforms the strongest model GPT-4o by +1.67%, while invoking GPT-4o in only 44.79% of samples. It also surpasses Qwen2.5-Max (+7.71%) and the reasoning-based DeepSeek-R1 (+4.17%). Furthermore, RAGRouter significantly outperforms non–RAG-aware baselines such as KNN Router (+4.17%) and MF (+2.92%). These results demonstrate that RAGRouter can effectively integrate closed-source models, leveraging their complementarity to achieve simultaneous improvements in both performance and efficiency.

Table 3: Testing accuracy (%) of RAGRouter and baselines with the inclusion of closed-source LLMs (Qwen2.5-Max, GPT-4o, and DeepSeek-R1) on NQ and WebQ, **bold** indicates best results.

| Method | NQ | WQ | Avg |
|---|---|---|---|
| Qwen2.5-32B-Instruct | 44.58 | 50.42 | 47.50 |
| Qwen2.5-72B-Instruct | 40.00 | 48.75 | 44.38 |
| Llama-3.3-70B-Instruct | 51.67 | 50.42 | 51.05 |
| Qwen2.5-Max | 51.25 | 52.08 | 51.67 |
| GPT-4o | 58.33 | 57.08 | 57.71 |
| DeepSeek-R1 | 56.67 | 53.75 | 55.21 |
| KNN Router | 58.75 | 51.67 | 55.21 |
| MF | 57.50 | 55.42 | 56.46 |
| **RAGRouter** | **59.58** | **59.17** | **59.38** |

## 5.4 Extended Results on Latency-Aware Routing

**Metrics.** Three metrics are used for evaluate latency-aware routing: **Area**, which measures the proportion of the area under the accuracy–latency curve within 1 second, reflecting overall time-efficiency; **Peak Acc**, the highest accuracy achieved within 1 second; and **Latency Gap-to-Match**, defined as the latency of the best-performing single LLM minus the minimum latency required by a method to match its accuracy, indicating the efficiency margin obtained through routing.

Table 4: Area (%), Peak Accuracy (PA, %), and Latency Gap-to-Match (G, s) for RAGRouter and baselines using score-threshold-based routing on MedMCQA (Local/Online) and TriviaQA. "–" indicates failure to match the best single-LLM performance; **bold** denotes the best result.

| Method | MedMCQA (Local) | | | MedMCQA (Online) | | | TQA | | |
|---|---|---|---|---|---|---|---|---|---|
| | Area ↑ | PA ↑ | G (s) ↑ | Area ↑ | PA ↑ | G (s) ↑ | Area ↑ | PA ↑ | G (s) ↑ |
| RouterDC | 55.87 | 62.22 | - | 57.77 | 65.56 | - | 70.99 | 75.83 | - |
| MF | 46.42 | 59.63 | 0.10 | 54.85 | 65.56 | 0.45 | 66.84 | 80.00 | - |
| GraphRouter | 47.96 | 59.30 | 0.18 | 57.82 | 64.67 | 0.32 | 65.42 | 73.15 | 0.01 |
| KNN Router | 52.26 | 62.30 | - | 60.18 | 66.10 | 0.33 | 72.61 | 81.65 | - |
| **RAGRouter** | **57.12** | **62.59** | **0.24** | **63.12** | **67.78** | **0.51** | **73.78** | **87.50** | **0.76** |

**Quantitative Results and Visualization.** We apply the score-threshold-based routing mechanism to RAGRouter and baselines across all tasks. Results on MedMCQA (Local/Online) and TriviaQA are reported in Table 4 and Figure 4, with supplementary results in Appendix C. RAGRouter consistently achieves the highest Area and Peak Accuracy across all tasks. Specifically, it outperforms baselines

---

[5] https://chat.qwen.ai
[6] https://platform.openai.com/docs/models/gpt-4o
[7] https://www.deepseek.com

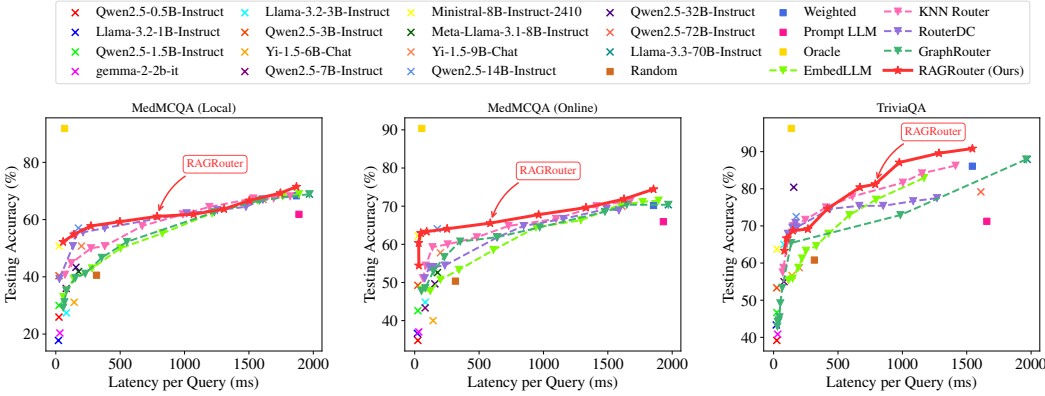

Figure 4: Accuracy–latency curves on MedMCQA (Local), MedMCQA (Online), and TriviaQA.

by 4.86%–10.7%, 2.94%–8.27%, and 1.17%–8.36% in Area, and by 0.29%–3.29%, 1.68%–3.11%, and 5.85%–14.35% in Peak Accuracy on MedMCQA (Local/Online) and TriviaQA, respectively. As shown in Figure 4, its accuracy–latency curve consistently dominates those of baselines, indicating superior accuracy under low-latency constraints. RAGRouter also achieves the largest positive Latency Gap-to-Match margins, demonstrating it can match the accuracy of the best single LLM with substantially lower latency. These results highlight RAGRouter's ability to exploit the optimization space enabled by retrieval augmentation, efficiently leveraging smaller, faster models to achieve the performance of larger ones without incurring unnecessary latency.

## 5.5 Sensitivity Analysis and Ablation Study

Table 5: Ablation study of Cross Encoder and RAG Capability Embedding Layer.

| Configuration | PopQA | | MedMCQA | | NQ | WQ | TQA | Avg | Δ |
|---|---|---|---|---|---|---|---|---|---|
| | Local | Online | Local | Online | | | | | |
| RAGRouter | 48.52 | 52.59 | 71.48 | 74.44 | 56.67 | 56.67 | 90.83 | 64.46 | 0.00 |
| w/o Cross Encoder | 47.78 | 52.22 | 70.74 | 74.44 | 55.42 | 55.00 | 88.75 | 63.48 | -0.98 |
| w/o RAG Capability Embedding Layer | 48.52 | 52.22 | 71.11 | 74.44 | 55.42 | 54.17 | 90.00 | 63.70 | -0.76 |

Table 6: Ablation study of contrastive learning objectives.

| ISC | CSC | PopQA | | MedMCQA | | NQ | WQ | TQA | Avg | Δ |
|---|---|---|---|---|---|---|---|---|---|---|
| | | Local | Online | Local | Online | | | | | |
| ✓ | ✓ | 48.52 | 52.59 | 71.48 | 74.44 | 56.67 | 56.67 | 90.83 | 64.46 | 0.00 |
| ✓ | ✗ | 48.15 | 52.22 | 71.11 | 73.33 | 53.75 | 56.25 | 89.58 | 63.49 | -0.97 |
| ✗ | ✓ | 48.52 | 51.85 | 71.48 | 74.07 | 55.00 | 56.25 | 89.58 | 63.82 | -0.64 |
| ✗ | ✗ | 47.78 | 51.48 | 69.26 | 70.74 | 53.75 | 53.75 | 89.17 | 62.28 | -2.18 |

**Ablation Study on RAGRouter Architecture.** We conduct an ablation study to evaluate the contributions of the Cross Encoder and the RAG Capability Embedding Layer in RAGRouter. As shown in Table 5, removing the Cross Encoder reduces performance by 0.98%, highlighting the importance of query-document interactions for deriving fused knowledge representations. Removing the RAG Capability Embedding Layer reduces performance by 0.76%; in this configuration, we removed the explicit modeling of RAG capabilities and instead assigned each LLM a single embedding capturing its overall behavior, without distinguishing between parametric knowledge and RAG capabilities. This result demonstrates that explicitly modeling RAG capabilities is essential for effective routing that accounts for each LLM's capacity to exploit external information.

**Effects of Positive and Negative Sample Selection in Contrastive Learning.** To assess the effectiveness of contrastive learning, we perform an ablation study on the two positive–negative sample construction strategies used in our method. As shown in Table 6, removing either Intra-Setting Contrast or Cross-Setting Contrast results in performance drops of 0.64% and 0.97%, respectively. When both components are removed—effectively disabling contrastive learning—accuracy decreases

Table 7: Effects of different candidate LLMs sets.

| Candidate Set | Method | PopQA | | MedMCQA | | NQ | WQ | TQA | Avg | Δ |
|---|---|---|---|---|---|---|---|---|---|---|
| | | Local | Online | Local | Online | | | | | |
| Small | Oracle Single Best | 45.56 | 51.48 | 40.37 | 49.26 | 36.67 | 45.42 | 65.00 | 47.68 | 0.00 |
| | RAGRouter | 45.56 | 51.85 | 40.37 | 51.48 | 36.67 | 47.92 | 65.00 | 48.41 | +0.73 |
| Large | Oracle Single Best | 46.30 | 50.37 | 68.89 | 70.37 | 51.67 | 50.42 | 87.92 | 60.85 | 0.00 |
| | RAGRouter | 47.41 | 50.74 | 71.11 | 73.33 | 54.58 | 53.33 | 90.42 | 62.99 | +2.14 |
| Small & Large | Oracle Single Best | 46.30 | 51.48 | 68.89 | 70.37 | 51.67 | 50.42 | 87.92 | 61.01 | 0.00 |
| | RAGRouter | 48.15 | 52.22 | 71.48 | 73.33 | 53.33 | 55.00 | 89.58 | 63.30 | +2.29 |

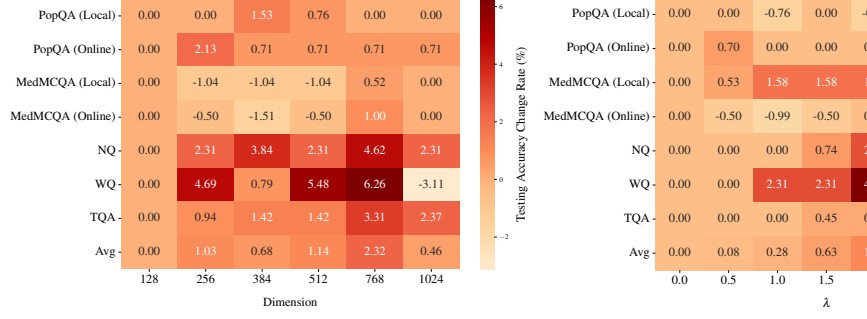

Figure 5: Effects of dimension (baseline: 128).    Figure 6: Effects of $\lambda$ (baseline: $\lambda = 0$).

by 2.18%. These findings underscore the importance of contrastive learning in capturing knowledge representation shifts and LLM heterogeneity, which is essential for effective routing in RAG settings.

**Effects of the Dimension of LLM Knowledge and RAG Capability Embeddings.** We investigate the impact of the dimensionality of both knowledge representation and RAG capability vectors. As shown in Figure 5, we observe that performance improves as the dimensionality increases, peaking at 768, after which it declines. Accordingly, we adopt a dimensionality of 768 in the main experiments. Detailed results are provided in Appendix D.

**Effects of $\lambda$.** We investigate the impact of the classification loss weight $\lambda$ in the overall loss function (Eq. 9) on test accuracy. As shown in Figure 6, we observe that combining contrastive and classification losses yields better performance than using contrastive loss alone (i.e., $\lambda = 0$, achieving only 63.76% on average). As $\lambda$ increases, accuracy improves, peaking at $\lambda = 2$ with 64.46% on average, before experiencing a slight decline. Based on this, we set $\lambda = 2$ in all experiments. Detailed results are provided in Appendix D.

**Effects of Different Candidate LLMs Sets.** We investigate how the composition of candidate LLMs affects routing performance. To this end, we form two subsets of models—Small ($\leq$3B parameters) and Large ($\geq$32B)—and evaluate three configurations: Small only, Large only, and a heterogeneous set combining both. As shown in Table 7, RAGRouter consistently outperforms the Oracle Single Best in all settings, demonstrating its ability to coordinate models and achieve cumulative gains. Two key insights emerge. First, the routing upper bound is largely determined by model strength: the Large set yields a significantly higher Oracle Single Best average (60.85%) than the Small set (47.68%), with RAGRouter following the same trend (62.99% vs. 48.41%). Second, combining heterogeneous models yields the best performance: the Small & Large setting achieves the highest RAGRouter average (63.30%) and the largest gain over its Oracle Single Best (+2.29%), suggesting that model diversity improves complementarity and enables more effective routing.

## 6  Conclusion

In this paper, we have studied the problem of LLM routing in Retrieval-Augmented Generation (RAG) for the first time and propose RAGRouter, the first RAG-aware routing method. By leveraging contrastive learning, RAGRouter captures knowledge representation shifts induced by external documents, enabling effective routing decisions. Experiments on diverse knowledge-intensive tasks demonstrate that RAGRouter outperforms existing non-RAG-aware methods and achieves strong performance-efficiency trade-offs under low-latency constraints.

## Acknowledgments and Disclosure of Funding

This work was supported in part by National Key R&D Program of China (No. 2022ZD0119100), China NSF grant No. 62025204, No. 62202296, No. 62272293, No. 62441236, No. U24A20326, and No. 62572299, Tencent WeChat Research Program, and SJTU-Huawei Explore X Gift Fund. The opinions, findings, conclusions, and recommendations expressed in this paper are those of the authors and do not necessarily reflect the views of the funding agencies or the government.

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

# Appendix Contents

# A  Experimental Setup Details

## A.1  Details of Candidate LLMs

The responses of Qwen2.5-72B-Instruct, Llama-3-70B-Instruct and closed-source LLMs were obtained via API calls, while other open-source LLMs were locally deployed using the vLLM framework [26] for high-speed inference from Huggingface[8]. Notably, latency calculations for API-based models incorporated both average network latency and inference time, whereas latency measurements for locally deployed models exclusively accounted for inference time.

## A.2  Details of Datasets

Table 8: The statistics results for different tasks.

| Query Num | PopQA | | MedMCQA | | NQ | WQ | TQA |
|---|---|---|---|---|---|---|---|
| | Local | Online | Local | Online | | | |
| Train | 2000 | 2000 | 2000 | 2000 | 1000 | 1000 | 1000 |
| Test | 270 | 270 | 270 | 270 | 240 | 240 | 240 |

As shown in Table 8, we randomly sampled queries from five knowledge-intensiv tasks (PopQA [32], MedMCQA [34], NQ [25], WebQ [2], and TriviaQA [23]) and partitioned them into training and test sets. For PopQA and MedMCQA, we retrieved documents through both local and online search engines following [42], while for NQ, WebQ, and TriviaQA, we constructed artificially noised documents based on [11], with an equal proportion of four types: Golden Context, Relevant Retrieval Noise, Irrelevant Retrieval Noise, and Counterfactual Retrieval Noise. Each query-document pair was processed by the 15 LLMs described in Section 5.1 to generate responses. These responses were then evaluated against ground-truth answers to derive binary scores.

# B  Impact of RAG on LLM Performance

**Performance Shift with and without RAG.** We analyze how LLM performance changes before and after RAG across different models and settings. As shown in Figure 7, on real retrieval settings such as PopQA (Online), MedMCQA (Local), and MedMCQA (Online), RAG improves overall performance and reduces the accuracy gap among models. Notably, small-scale models benefit more (e.g., Qwen2.5-0.5B-Instruct achieves a 17.57% improvement on MedMCQA (Online)) compared to larger models (e.g., Llama-3.3-70B-Instruct improves by only 0.88%). In contrast, under noisy retrieval settings (NQ, WebQ, TriviaQA), the impact of RAG varies—some models improve while others degrade (e.g., on NQ, Llama-3.3-70B-Instruct performs worse, while Qwen2.5-0.5B-Instruct performs better). These results highlight inconsistent performance shifts across models, indicating that RAG significantly alters the distribution of response quality, which undermines the assumptions of non-RAG-aware routing strategies.

**Analysis of Response Quality Reversal.** We further investigate how RAG causes quality reversals for the same query. To quantify these reversals, we define two metrics at the task level. The positive gain rate is the proportion of queries that were unanswerable without retrieval but became answerable with retrieved documents, calculated as:

$$\text{Positive Gain Rate} = \frac{\#(\text{Incorrect w/o RAG} \rightarrow \text{Correct w/ RAG})}{\#(\text{Incorrect w/o RAG})} \tag{10}$$

The negative interference rate measures the opposite effect—queries that were answerable without retrieval but became unanswerable due to retrieved content:

$$\text{Negative Interference Rate} = \frac{\#(\text{Correct w/o RAG} \rightarrow \text{Incorrect w/ RAG})}{\#(\text{Correct w/o RAG})} \tag{11}$$

Figures 8 and 9 report these metrics across 15 LLMs and multiple settings. On average, across all models and tasks, the positive gain rate is 30.86%, negative interference rate is 31.46%. These results confirm that response quality reversals induced by external documents are common suggests that RAG-induced knowledge shifts are widespread among LLMs.

---

[8]`https://huggingface.co`

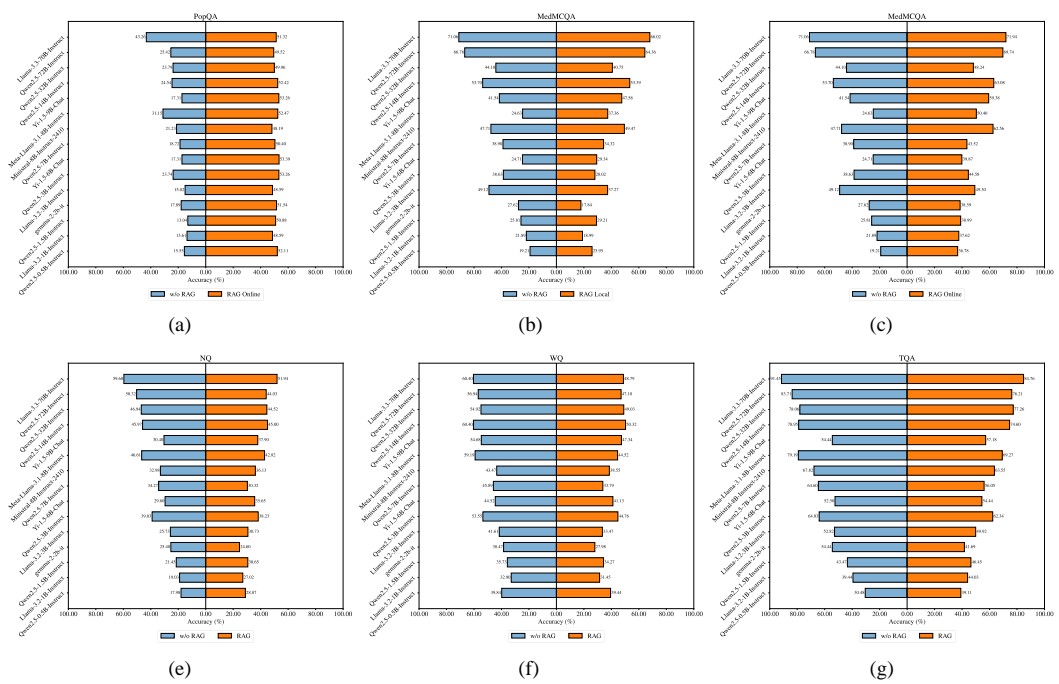

Figure 7: Accuracy of 15 LLMs before and after RAG under various tasks and retrieval settings: (a) PopQA (Online), (b) MedMCQA (Local), (c) MedMCQA (Online), (d) NQ, (e) WebQ, (f) TQA.

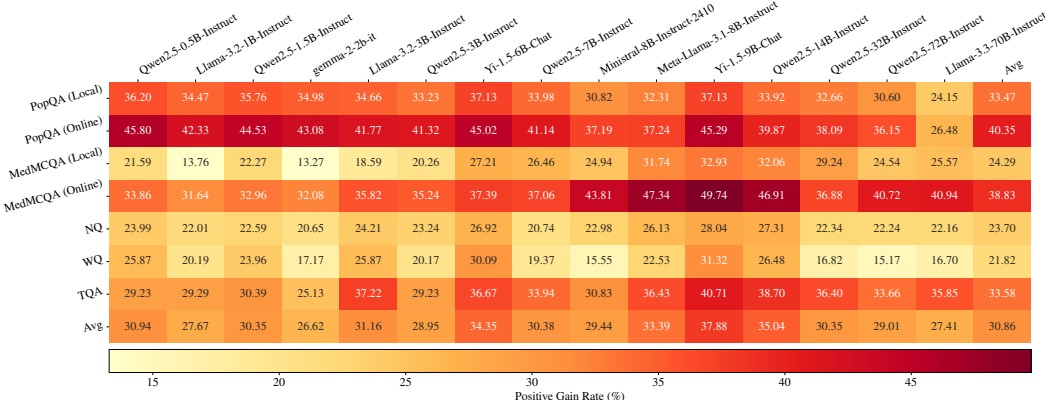

Figure 8: Positive Gain Rates of 15 candidate LLMs across tasks.

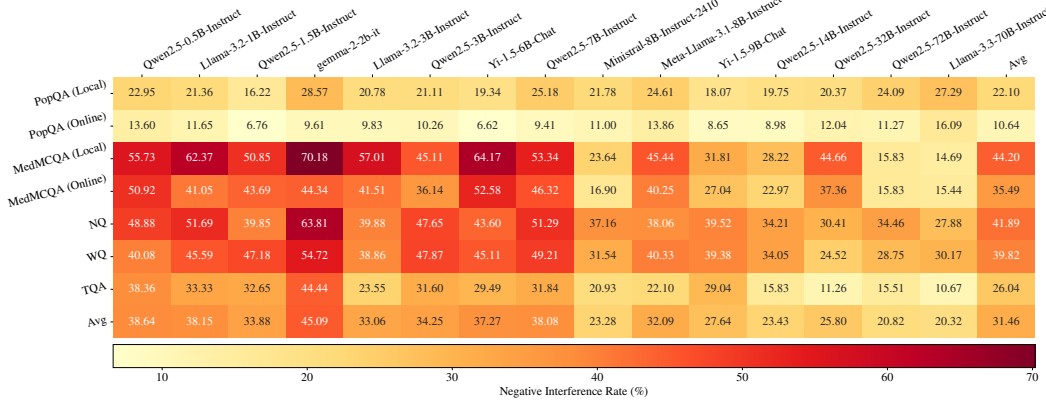

Figure 9: Negative Interference Rates of 15 candidate LLMs across tasks.

# C Full Results on Latency-Aware Routing

**Setting Details.** Based on LLMs' profiles, the 15 candidate models in Section 5.4 were ranked as follows in ascending order: Qwen2.5-0.5B-Instruct, Llama-3.2-1B-Instruct, Qwen2.5-1.5B-Instruct, gemma-2-2b-it, Llama-3.2-3B-Instruct, Qwen2.5-3B-Instruct, Yi-1.5-6B-Chat, Qwen2.5-7B-Instruct, Ministral-8B-Instruct-2410, Meta-Llama-3.1-8B-Instruct, Yi-1.5-9B-Chat, Qwen2.5-14B-Instruct, Qwen2.5-32B-Instruct, Qwen2.5-72B-Instruct, Llama-3.3-70B-Instruct. Substitution models were selected from immediate predecessors of the highest-routing-score model within the threshold. For quantitative analysis, we discretized the threshold parameter $\theta$ over [0, 1] with a step size of 1e-4 to generate complete high-precision accuracy–latency trade-off curves.

**Results.** Figure 10 illustrates the accuracy–latency curves of RAGRouter and baseline methods on PopQA (Local/Online), NQ, and WebQ, while Tables 9 present their quantitative results on Area, Peak Acc, and Latency Gap-to-Match metrics. RAGRouter achieves the highest scores in Area and Peak Acc, with its accuracy–latency curve mostly surpassing the baselines, demonstrating strong performance-efficiency trade-offs under low-latency constraints.

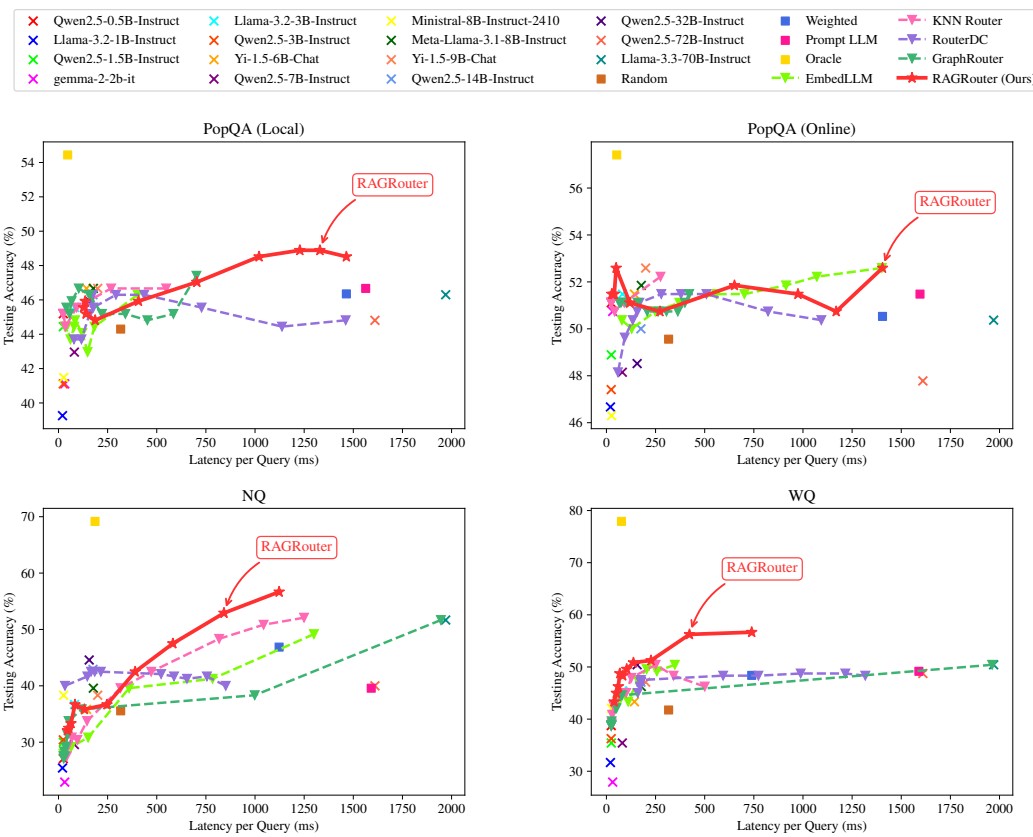

Figure 10: Accuracy–latency curves on PopQA (Local/Online), NQ and WebQ.

Table 9: Area (%), Peak Accuracy (PA, %), and Latency Gap-to-Match (G, s) for RAGRouter and baselines using score-threshold-based routing on PopQA (Local/Online), NQ and WebQ. "-" in Latency Gap-to-Match indicates failure to match the best single-LLM performance; "-" in Area denotes maximum routing latency below 1s, excluded from comparison; **bold** denotes the best result.

| Method | PopQA (Local) | | | PopQA (Online) | | | NQ | | | WQ | | |
|---|---|---|---|---|---|---|---|---|---|---|---|---|
| | Area ↑ | PA ↑ | G (s) ↑ | Area ↑ | PA ↑ | G (s) ↑ | Area ↑ | PA ↑ | G (s) ↑ | Area ↑ | PA ↑ | G (s) ↑ |
| RouterDC | 44.58 | 47.04 | **0.02** | 49.80 | 51.85 | - | 33.10 | 42.92 | - | 46.39 | 49.17 | - |
| MF | - | 46.30 | - | 49.96 | 52.22 | -0.91 | 37.34 | 44.17 | - | - | 50.42 | 1.66 |
| GraphRouter | - | 47.41 | -0.50 | - | 51.48 | - | 35.68 | 38.35 | 0.02 | 45.67 | 48.18 | 0.01 |
| KNN Router | - | 46.67 | -0.06 | - | 52.22 | - | 41.13 | 50.63 | 0.72 | - | 50.42 | 1.72 |
| RAGRouter | **45.13** | **48.52** | -0.48 | **50.13** | **52.59** | 0.15 | **43.63** | **55.83** | 1.13 | - | **58.33** | **1.84** |

# D    Full Results of Sensitivity Analysis and Ablation Study

Table 10 presents the impact of the dimensionality of LLM knowledge embeddings and RAG capability embeddings on test accuracy, the best average accuracy is observed at dimension 768.

Table 10: Effects of dimension.

| Dimension | PopQA | | MedMCQA | | NQ | WQ | TQA | Avg |
|---|---|---|---|---|---|---|---|---|
| | Local | Online | Local | Online | | | | |
| 128 | 48.52 | 52.22 | 71.11 | 73.70 | 54.17 | 53.33 | 87.92 | 63.00 |
| 256 | 48.52 | 53.33 | 70.37 | 73.33 | 55.42 | 55.83 | 88.75 | 63.65 |
| 384 | 49.26 | 52.59 | 70.37 | 72.59 | 56.25 | 53.75 | 89.17 | 63.43 |
| 512 | 48.89 | 52.59 | 70.37 | 73.33 | 55.42 | 56.25 | 89.17 | 63.72 |
| 768 | 48.52 | 52.59 | 71.48 | 74.44 | 56.67 | 56.67 | 90.83 | 64.46 |
| 1024 | 48.52 | 52.59 | 71.11 | 73.70 | 55.42 | 51.67 | 90.00 | 63.29 |

Table 11 presents the impact of the loss weight $\lambda$ on test accuracy, the best average accuracy is observed at $\lambda = 2$.

Table 11: Effects of $\lambda$.

| $\lambda$ | PopQA | | MedMCQA | | NQ | WQ | TQA | Avg |
|---|---|---|---|---|---|---|---|---|
| | Local | Online | Local | Online | | | | |
| 0.0 | 48.89 | 52.59 | 70.37 | 74.44 | 55.42 | 54.17 | 90.42 | 63.76 |
| 0.5 | 48.89 | 52.96 | 70.74 | 74.07 | 55.42 | 54.17 | 90.42 | 63.81 |
| 1.0 | 48.52 | 52.59 | 71.48 | 73.70 | 55.42 | 55.42 | 90.42 | 63.94 |
| 1.5 | 48.89 | 52.59 | 71.48 | 74.07 | 55.83 | 55.42 | 90.83 | 64.16 |
| 2.0 | 48.52 | 52.59 | 71.48 | 74.44 | 56.67 | 56.67 | 90.83 | 64.46 |
| 2.5 | 48.52 | 52.96 | 71.48 | 74.44 | 57.08 | 56.25 | 90.42 | 64.45 |
| 3.0 | 48.52 | 52.96 | 71.85 | 74.44 | 55.83 | 55.83 | 90.00 | 64.21 |

# E    Case Studies of Failures and Successes

## E.1    Quantitative Failure Case Studies of Routing Decisions

We conduct a quantitative analysis of cases where RAGRouter and non-RAG-aware routing methods fail to select the correct LLM or route correctly on the PopQA (Local), MedMCQA (Local), and NQ datasets. In particular, we categorize non-trivially unsolvable failures (i.e., excluding cases where all LLMs fail) into four types as follows:

- F1: Failure to perceive performance degradation caused by RAG (i.e., cases where LLM's performance decreases after RAG is applied).

- F2: Inherent task difficulty, where the majority of LLMs (e.g., >80%) fail, leading to routing confusion.

- F3: Overconfident selection of stronger LLMs for cases where high-capacity LLMs are chosen but still fail, outside the conditions of F1 and F2.

- F4: Other factors, such as outliers or ambiguous inputs.

Table 12: Type-wise failure rates (as a percentage of all cases) and the overall failure rate (FR).

| Method | F1 (%) | F2 (%) | F3 (%) | F4 (%) | FR (%) |
|---|---|---|---|---|---|
| KNN Router | 5.00 | 8.33 | **1.28** | 1.54 | 16.15 |
| MF | 5.51 | 8.21 | 1.41 | 1.79 | 16.92 |
| **RAGRouter** | **3.85** | **7.05** | 1.41 | **0.64** | **12.95** |

As shown in Table 12, we can observe that compared with non-RAG-aware routing methods, including KNN Router and MF, our RAGRouter achieves the lowest overall failure rate and the lowest F1-type failure rate. This suggests that RAGRouter more correctly captures performance shifts in LLMs introduced by RAG.

We additionally analyze cases where RAGRouter fails while non-RAG-aware routing methods succeed. We find that such cases are less than 2% of all the evaluated cases. Notably, more than 50% of these cases fall into the F2 type, where the performance gap between candidate LLMs is inherently small.

## E.2 Qualitative Success Case Studies of RAGRouter

We further illustrate RAGRouter's ability to perceive changes in LLM knowledge states under RAG conditions through the two case studies shown in Table 13 and Table 14. In Table 13, the retrieved document contains both correct answer information and certain distracting content. In this case, RAGRouter identifies that the document provides significant performance gains for Qwen2.5-14B-Instruct and accordingly selects it as the responder. Although this model produces an incorrect response in the non-RAG setting due to confusion between two French departments, it successfully corrects the answer when the document is incorporated. In contrast, Meta-Llama-3.1-8B-Instruct exhibits greater sensitivity to distracting content in the document, where the introduction of RAG leads to reverse interference and impairs its reasoning, and thus it is not prioritized by RAGRouter. However, traditional routing strategies without RAG awareness, such as MF, rely solely on static model performance and fail to perceive the document-induced performance shifts, ultimately routing to Meta-Llama-3.1-8B-Instruct and resulting in routing failure for this sample.

Table 13: A case from PopQA (Online), demonstrating RAGRouter's ability to perceive LLMs' RAG capability and knowledge shift.

| Query | What is Agen the capital of? | |
|---|---|---|
| Document | Agen, located in the Nouvelle-Aquitaine region of Southwestern France, serves as the prefecture of the Lot-et-Garonne department. Known for its geographical positioning along the river Garonne, the city lies approximately 135 kilometers southeast of Bordeaux. It has a rich cultural heritage, featuring various historical buildings such as the twelfth-century Agen Cathedral and numerous museums, including the Musée des Beaux Arts. Agen is also colloquially referred to as the "capital of the prune," hosting a popular prune festival every August. The town, with a population of 32,485 in 2021, has its own Roman Catholic diocese, adding to its historical significance within the region. | |
| Ground truth | Lot-et-Garonne | |
| Router | RAGRouter | MF |
| Selected LLM | Qwen2.5-14B-Instruct | Meta-Llama-3.1-8B-Instruct |
| Response w/o RAG | Lot department. (✗) | Lot-et-Garonne department. (✓) |
| Response w/ RAG | Lot-et-Garonne department. (✓) | The prune capital. (✗) |

As shown in Table 14, the query itself is challenging, and neither Llama-3.3-70B-Instruct nor Qwen2.5-72B-Instruct is able to provide a correct answer without access to external documents. However, when the retrieved document containing key information is provided, Llama-3.3-70B-Instruct, benefiting from its stronger capabilities in information extraction and comprehension, successfully identifies "Poreotics" as the correct answer. In contrast, Qwen2.5-72B-Instruct fails to effectively utilize the document and still produces an incorrect response. In this scenario, RAGRouter is able to sense the differential capabilities of the candidate LLMs in leveraging retrieved content and routes the query to the model with stronger information extraction ability, leading to a correct response. By contrast, non-RAG-aware routing methods based on static modeling assumptions, such as RouterDC, fail to capture dynamic performance shifts induced by retrieval and result in incorrect routing and response failure. This case further highlights the advantage of RAGRouter in perceiving capability differences among LLMs in retrieval-augmented settings.

Table 14: A case from NQ, demonstrating RAGRouter's ability to perceive LLMs' RAG capability and knowledge shift.

| Query | who are the dancers in the lazy song video? | |
|---|---|---|
| Document | tenth was the one chosen. The official video was directed by Mars and Cameron Duddy, produced by Nick Tabri and Dara Siegel, and features Poreotics wearing chimpanzee masks; it was released on April 15, 2011. The whole video is presented in as a lone continuous and uninterrupted shot, it begins with Mars singing and hanging out in a bedroom with five dancers, they all wear monkey masks and Mars dresses in black sunglasses and a flannel shirt. While Mars sings what he feels to do on a day off, he and the monkeys perform dance moves typical of a boy-band, | |
| Ground truth | Poreotics | |
| Router | RAGRouter | RouterDC |
| Selected LLM | Llama-3.3-70B-Instruct | Qwen2.5-72B-Instruct |
| Response w/o RAG | Bruno Mars and dancers. (✗) | Five dancers. (✗) |
| Response w/ RAG | Poreotics dancers. (✓) | Bruno Mars and his backup dancers. (✗) |

# F  Routing Performance under Cross-Domain Settings

To evaluate RAGRouter's generalization across domains, we design two cross-domain settings:

- Setting 1: Trained on MedMCQA (Local) that falls into medical domain, but tested on a new dataset HotpotQA [54] for Wikipedia-based multi-hop QA.
- Setting 2: Trained on NQ that contains real-world search queries, but tested on MedMCQA (Local).

Table 15: Testing accuracy (%) of RAGRouter and baselines under two cross-domain settings.

| Method | Setting 1
MedMCQA (Local)→HotpotQA | Setting 2
NQ→MedMCQA (Local) |
|---|---|---|
| KNN Router | 18.60 | 63.33 |
| MF | 20.20 | 58.89 |
| **RAGRouter** | **21.20** | **68.89** |

As shown in Table 15, RAGRouter outperforms non–RAG-aware routing methods by 1.00% and 5.56% in two cross-domain settings, demonstrating strong generalization. This performance advantage can be attributed to RAGRouter's contrastive learning framework, which effectively captures relative RAG capability differences among candidate LLMs, even when transferred across domains.

# G  Routing Performance under Noisy Retrieval

We further partition the TriviaQA dataset into four subsets with manually injected retrieval noise—Golden Context, Relevant Noise, Irrelevant Noise, and Counterfactual Noise—to evaluate the routing effectiveness of RAGRouter under different noise conditions, in comparison with various baseline methods. As shown in Table 16, RAGRouter consistently achieves the best performance across all subsets, outperforming both the Oracle Single Best and other routing strategies, demonstrating strong robustness to different types of retrieval noise.

Notably, on the Relevant Noise, Irrelevant Noise, and Counterfactual Noise subsets, non-RAG-aware baselines exhibit significant performance gaps compared to RAGRouter, highlighting their limitations in high-noise retrieval scenarios. We hypothesize that this is due to knowledge shift induced by noisy retrieval, which affects different LLMs in heterogeneous ways. As a result, routing methods that rely on fixed model representations and ignore RAG capabilities struggle to accurately model routing strategies under such conditions.

Table 16: Test accuracy (%) of RAGRouter and baselines on TriviaQA subsets with different types of retrieval noise, **bold** indicates best results.

| Method | Golden Context | Relevant Noise | Irrelevant Noise | Counterfactual Noise | Avg |
|---|---|---|---|---|---|
| Oracle Single Best | 95.00 | 83.33 | **90.00** | 83.33 | 87.92 |
| KNN Router | 96.67 | 78.33 | 83.33 | 86.67 | 86.25 |
| GraphRouter | 95.00 | 83.33 | **90.00** | 83.33 | 87.92 |
| RouterDC | 80.00 | 78.33 | 73.33 | 78.33 | 77.50 |
| MF | 95.00 | 76.67 | 80.00 | 80.00 | 82.92 |
| **RAGRouter** | **98.33** | **86.67** | **90.00** | **88.33** | **90.83** |

## H  Routing Performance on Summarization Task

To verify that RAGRouter is also applicable to RAG tasks beyond question answering, we conduct experiments on WikiASP [16], a summarization task commonly used in RAG-related research. Specifically, WikiASP aims to generate aspect-based summaries for Wikipedia entities and naturally supports the use of retrieved evidence. We adopt ROUGE-L as the evaluation metric.

For continuous performance metrics such as ROUGE-L, we extend RAGRouter's contrastive learning by score-based probabilistic sampling. Given each candidate LLM's normalized score $s \in [0, 1]$, we label it as "can answer" (positive, label 1) with probability $s$, and "cannot answer" (negative, label 0) with probability $1 - s$. This strategy is applied in our WikiASP experiments, enabling contrastive training based on continuous performance signals rather than discrete correctness labels.

Table 17: Performance of RAGRouter and baselines on WikiASP measured by ROUGE-L.

| Method | WikiASP |
|---|---|
| Qwen2.5-0.5B-Instruct | 0.1388 |
| Qwen2.5-1.5B-Instruct | 0.1411 |
| Llama-3.2-3B-Instruct | 0.1768 |
| Qwen2.5-3B-Instruct | 0.1528 |
| Qwen2.5-7B-Instruct | 0.1897 |
| Llama-3.1-8B-Instruct | 0.1455 |
| KNN Router | 0.1881 |
| MF | 0.1865 |
| **RAGRouter** | **0.1981** |

As shown in Table 17, RAGRouter outperforms non-RAG-aware routing baselines by more than 5.3%. These findings demonstrate that RAGRouter generalizes well to different RAG tasks.

