# OpenReview forum: "RAGRouter: Learning to Route Queries to Multiple Retrieval-Augmented Language Models"
_NeurIPS.cc/2025/Conference — NeurIPS 2025 poster_

### Official Review · Reviewer_WwUD · 2025-06-13

**Clarity:** 2
**Significance:** 3
**Originality:** 2
**Rating:** 4
**Confidence:** 3

**Summary:**

The paper titled "RAGRouter: Learning to Route Queries to Multiple Retrieval-Augmented Language Models" introduces a novel routing framework, RAGRouter, designed to optimize the selection of the most suitable Large Language Model (LLM) for a given query in the context of Retrieval-Augmented Generation (RAG). The authors address the challenge of varying response quality across different LLMs when augmented with external knowledge, proposing a method that leverages document embeddings and contrastive learning to capture dynamic shifts in model capabilities. The results demonstrate that RAGRouter significantly outperforms individual LLMs and existing routing methods, achieving superior performance and efficiency trade-offs under low-latency constraints.

**Questions:**

1. How sensitive is RAGRouter to the quality of retrieved documents? In scenarios with poor retrieval, does the method degrade gracefully, or does it significantly impact performance?
2. Given the computational complexity of the training process, are there any plans to explore more efficient optimization strategies or lightweight architectures to improve scalability?
3. How does RAGRouter compare with other recent routing methods that also aim to capture dynamic shifts in LLM capabilities? Are there any specific advantages or trade-offs?

**Ethical Concerns:**

["NO or VERY MINOR ethics concerns only"]

**Limitations:**

1. While the method is novel in its application to RAG, it builds on existing techniques in contrastive learning and routing. Future work could explore more innovative approaches to capturing dynamic shifts in LLM capabilities.
2. The training process, particularly the contrastive learning component, requires significant computational resources. This may limit the adoption of RAGRouter in resource-constrained environments.

**Quality:**

3

**Strengths And Weaknesses:**

Strengths:
1. The paper presents a approach to routing queries in RAG settings, capturing dynamic shifts in LLM capabilities through contrastive learning. This method effectively addresses the limitations of static routing strategies.
2. The authors conduct extensive experiments across diverse knowledge-intensive tasks and retrieval settings, demonstrating that RAGRouter consistently outperforms both individual LLMs and existing routing methods.
3. The extended score-threshold-based mechanism allows RAGRouter to achieve strong performance-efficiency trade-offs, making it suitable for low-latency applications.

Weaknesses:
1. While the method is novel in its application to RAG, it builds on existing techniques in contrastive learning and routing. The innovation may be seen as incremental rather than groundbreaking.
2. The effectiveness of RAGRouter relies heavily on the quality and relevance of retrieved documents. In scenarios with poor retrieval, the performance gains may be diminished.

---

> ### Author Rebuttal · Authors · 2025-07-30
>
> Thank you for your thoughtful feedback. We have addressed your concerns in detail below.
>
> > **Q1**: Technical innovation.
>
> **A1**: We would like to emphasize that **our RAGRouter is not merely a combination of existing techniques in contrastive learning and routing, but a novel framework driven by a previously unaddressed problem: RAG-aware dynamic LLM routing.**
>
> **We are the first to explicitly define the new RAG-aware LLM routing problem.** As RAG becomes a mainstream paradigm, **different LLMs vary significantly in how they incorporate and utilize retrieved, non-parametric knowledge**. However, **existing LLM routing methods typically assume static model capabilities and overlook the impact of retrieval context.**
>
> To deal with this new problem, we design a new router network architecture that introduces **document encoder and RAG capability embedding layer** to model **how LLMs absorb and respond to retrieved evidence**. Further considering **the retrieval setting can substantially alter an LLM’s answerability by shifting the knowledge representation**, we propose **intra- and cross-setting contrastive learning objectives to capture the performance variations with and without RAG settings**. Ablation studies have also validated the necessity and effectiveness of each design module.
>
> ---
>
> > **Q2**: Performance gains in scenarios with poor retrieval.
>
> **A2**: **We have reported and analyzed RAGRouter’s performance under noisy retrieved documents in Appendix C (Routing Performance under Noisy Retrieval) of the submission.** In particular, we add synthetic noise into the NQ, WebQ, and TriviaQA datasets. Take TriviaQA for example. The evaluation results under four conditions, including Golden Context, Relevant Noise, Irrelevant Noise, and Counterfactual Noise, are summarized as follows.
>
> |        Method        | Golden Context | Relevant Noise | Irrelevant Noise | Counterfactual Noise |    Avg    |
> | :------------------: | :------------: | :------------: | :--------------: | :------------------: | :-------: |
> |      KNN Router      |     96.67      |     78.33      |      83.33       |        86.67         |   86.25   |
> |          MF          |     95.00      |     76.67      |      80.00       |        80.00         |   82.92   |
> | **RAGRouter (Ours)** |   **98.33**    |   **86.67**    |    **90.00**     |      **88.33**       | **90.83** |
>
> We can observe that **our RAGRouter consistently outperforms traditional LLM routing methods, with performance gains more than 4.58%, highlighting its strong robustness to retrieval noise**.
>
> ---
>
> > **Q3**: Comparison with recent routing methods that also aim to capture dynamic shifts in LLM capabilities.
>
> **A3**: We have conducted a comprehensive literature search and found some concurrent works [1,2] that also explore routing strategies under dynamically shifting LLM capabilities. **These studies** typically focus on reasoning-oriented scenarios and capture **the internal knowledge of LLMs** (e.g., different CoT strategies and internal state transitions) to guide routing decisions.
>
> In contrast, **our RAGRouter** focuses on a totally different setting: routing for **LLM with RAG**, where **capability shifts arise from the interaction between LLMs and external knowledge.** This introduces new factors, including varying document quality, model-specific sensitivities to retrieved content, and compatibility between models and documents, which were not found in existing reasoning-centric scenarios. RAGRouter models these new factors to enable effective routing in RAG.  It captures **how different LLMs respond to varying retrieved contexts through a new network architecture and contrastive learning framework**. In contrast, **existing LLM routing methods focus on reasoning dynamics and cannot represent or utilize external knowledge**, making them **not applicable for the new RAG routing setting.**
>
> In summary, our RAGRouter can be viewed as a complementary and parallel approach to existing LLM routing methods, offering specific advantages in RAG scenarios, where external knowledge plays a central role in shaping model behavior.
>
> [1] Pan Z, Zhang K, Zhao Y, Han Y. Route to Reason: Adaptive Routing for LLM and Reasoning Strategy Selection. arXiv preprint arXiv:2505.19435. May 26, 2025.
>
> [2] Zhang W, Qiao S, Luo L, Li Y, Zheng C, Xu Q, Li M, Gui Y, He Y, Qiu J, Hong J. SynapseRoute: An Auto-Route Switching Framework on Dual-State Large Language Model. arXiv preprint arXiv:2507.02822. Jul 3, 2025.
>
> ---
>
> > **Q4**: Training efficiency of RAGRouter.
>
> **A4**: We clarify that **RAGRouter and all other LLM routers are trained on resource-rich cloud infrastructure**. Despite this, **RAGRouter remains lightweight, with just 136M parameters**. It was trained using **a single consumer-grade NVIDIA RTX 4090D GPU**, achieving an average training time of **11.6 minutes on PopQA and MedMCQA**, and **4.5 minutes on NQ, WebQ, and TQA**. The **peak memory usage** during training was only **18.9 GB**.

---

### Official Review · Reviewer_SuEE · 2025-06-16

**Clarity:** 3
**Significance:** 3
**Originality:** 3
**Rating:** 4
**Confidence:** 4

**Summary:**

This paper introduces RAGRouter, a novel framework for routing queries to the best language model in a Retrieval-Augmented Generation (RAG) setting. RAGRouter explicitly models three factors: (1) the semantics of retrieved documents, (2) query–document interaction via a cross-encoder, and (3) each model’s learned “RAG capability” embedding. Experiments on five knowledge-intensive QA benchmarks show a large performance improvement.

**Questions:**

1. How does the model perform on commercial strong LLMs, such as GPT4o?
2. How well does RAGRouter generalize to other RAG tasks?

**Ethical Concerns:**

["NO or VERY MINOR ethics concerns only"]

**Final Justification:**

I have read the rebuttal, which has fully resolved my concerns. I decide to keep my score positive.

**Limitations:**

See weakness above.

**Quality:**

3

**Strengths And Weaknesses:**

Strengths:
1. Solid empirical gains across five diverse QA datasets, with thorough ablations on each component.
2. The motivation is solid and easy to understand.
3. The proposed method is elegant and modularly presented.

Weaknesses:
1. The experiments seem to only consider open-source LLMs.
2. The evaluation is mainly conducted on QA, and it remains unclear how well RAGRouter generalizes to other RAG tasks (e.g., summarization, dialogue)
3. It requires ground-truth labels of “can answer vs. cannot answer”.

---

> ### Author Rebuttal · Authors · 2025-07-30
>
> Thanks for your thoughtful feedback on our paper. We have provided responses to all your concerns below.
>
> > **Q1**: Performance on commercial strong LLMs, such as GPT4o.
>
> **A1**: We have followed your suggestion to **add commercial strong LLMs, including GPT-4o and Qwen2.5-Max**, to the original candidate LLM pool with parameter size $\ge$ 32B. We then evaluate our RAGRouter on NQ and WebQ, comparing its performance against non–RAG-aware baselines, including KNN Router and MF.
>
> |           Method            |    NQ     |   WebQ    |    Avg    |
> | :-------------------------: | :-------: | :-------: | :-------: |
> |    Qwen2.5-32B-Instruct     |   44.58   |   50.42   |   47.50   |
> |    Qwen2.5-72B-Instruct     |   40.00   |   48.75   |   44.38   |
> |   Llama-3.3-70B-Instruct    |   51.67   |   50.42   |   51.05   |
> |         Qwen2.5-max         |   51.25   |   52.08   |   51.67   |
> | GPT-4o |   58.33   |   57.08   |   57.71   |
> |         KNN Router          |   57.08   |   50.42   |   53.75   |
> |             MF              |   57.08   |   55.83   |   56.46   |
> |    **RAGRouter (Ours)**     | **58.75** | **57.92** | **58.34** |
>
> Evaluation results show that although **GPT-4o is the top-performing individual model**, **our RAGRouter delivers an additional +0.63% improvement**, while **invoking GPT-4o in only 52.3% of samples**. **RAGRouter further outperforms Qwen2.5-Max (+6.67%), KNN Router (+4.59%), and MF (+1.88%)**. These results highlight RAGRouter’s effectiveness even in the presence of highly powerful LLMs, **leveraging model complementarity to surpass each single LLM, with better latency and cost efficiency by selectively using GPT-4o**.
>
> ---
>
> > **Q2**: How well does RAGRouter generalize to other RAG tasks?
>
> **A2**: Following your suggestion, we have supplemented experiments on **WikiASP [1], a summarization task commonly used in RAG-related research**. In particular, WikiASP aims to generate aspect-based summaries for Wikipedia entities and naturally supports the use of retrieved evidence. We adopt ROUGE-L as the evaluation metric.
>
> |        Method         |  WikiASP   |
> | :-------------------: | :--------: |
> | Qwen2.5-0.5B-Instruct |   0.1388   |
> | Qwen2.5-1.5B-Instruct |   0.1411   |
> | Llama-3.2-3B-Instruct |   0.1768   |
> |  Qwen2.5-3B-Instruct  |   0.1528   |
> |  Qwen2.5-7B-Instruct  |   0.1897   |
> | Llama-3.1-8B-Instruct |   0.1455   |
> |      KNN Router       |   0.1881   |
> |          MF           |   0.1865   |
> | **RAGRouter (Ours)**  | **0.1981** |
>
> Evaluation results show that **RAGRouter outperforms non-RAG-aware routing baselines by more than 5.3%**. These findings demonstrate that RAGRouter generalizes well to different RAG tasks.
>
> [1] Hayashi, Hiroaki, et al. "Wikiasp: A dataset for multi-domain aspect-based summarization." Transactions of the Association for Computational Linguistics 9 (2021): 211-225.
>
> ---
>
> > **Q3**: It requires ground-truth labels of "can answer vs. cannot answer".
>
> **A3**: We clarify that **RAGRouter doesn't rely on strict "can answer vs. cannot answer" discrete labels.** For **continuous performance metrics like ROUGE-L**, we can use a **score-based probabilistic sampling** strategy to generate positive and negative sample pairs for contrastive learning in our RAGRouter. **Specifically, for each candidate LLM's score $s\in[0,1]$, we label it as "can answer" (positive sample, label 1) with probability $s$, and "cannot answer" (negative sample, label 0) with probability $1-s$.** We **have applied this strategy in our newly added experiments on the WikiASP summarization task using ROUGE-L scores**, and the results validate its effectiveness. This further highlights the flexibility of RAGRouter in handling tasks with continuous performance metrics, without strictly requiring binary supervision.

---

> ### Comment · Reviewer_SuEE · 2025-08-06
>
> Thanks for your feedback! The rebuttal has fully resolved my questions and I decide to keep my score positive.

---

> > ### Author Response · Authors · 2025-08-06
> >
> > Thank you very much for your kind feedback. We're delighted to hear that our responses have fully addressed your concerns. We greatly appreciate your positive recognition and thoughtful review, and we will incorporate the suggested revisions into the final version.

---

### Official Review · Reviewer_s6eC · 2025-06-20

**Clarity:** 3
**Significance:** 3
**Originality:** 4
**Rating:** 4
**Confidence:** 4

**Summary:**

This paper proposes RAGRouter, a routing framework for retrieval-augmented generation (RAG) models that selects the most suitable model for each query by explicitly modeling knowledge shifts introduced by retrieval. It combines document embeddings and RAG-specific representations through contrastive learning, achieving 3.61% higher accuracy on average compared to the best single model and 3.29%–9.33% improvement over existing routing methods across five knowledge-intensive tasks.

**Questions:**

Will the dataset used to train RAGRouter be open-sourced? Releasing it would greatly promote progress in this area.

And please refer to the other questions in the weaknesses section.

**Ethical Concerns:**

["NO or VERY MINOR ethics concerns only"]

**Final Justification:**

All my raised issues are resolved. Meanwhile, authors supplemented my questions with experiments on reasoning-based models and powerful closed-source LLMs, so I will maintain my current positive rating.

**Limitations:**

yes

**Paper Formatting Concerns:**

no concerns.

**Quality:**

3

**Strengths And Weaknesses:**

Strength:
1. Writing is quite clear, and I can easily follow the author's idea and logic.
2. Code is open-sourced, it's really convenient for following research.
3. The figures and tables are very well-designed, and I really enjoyed reading this paper.

Weaknesses:
1. In lines 134–145, the authors mention three core factors, which motivate the design of different modules in lines 146–169. However, the correspondence between them is somewhat unclear. The authors are encouraged to clearly state the design motivation at the beginning of each paragraph in lines 146–169.
2. The experiments did not include very strong closed-source models (e.g., gpt4o) or reasoning-oriented models (e.g., deepseek-r1) as the LLMs. What is the reason for this? Is it possible that these models are powerful enough on their own and therefore do not need RAGRouter?
3. The current capability vector is set to a dimensionality of 768. Based on experience from recommender system literature, this dimensionality can significantly impact model performance. I suggest adding related ablation or hyperparameter experiments to study this effect.

---

> ### Author Rebuttal · Authors · 2025-07-30
>
> Thank you for your careful review and valuable feedback. We address all your comments below.
>
> > **Q1**: Clarify the correspondence between the three core RAG-aware factors and the design of different modules.
>
> **A1**: We clarify the correspondence as follows:
>
> - The factor **“the non-parametric knowledge provided by the documents”** corresponds to our **Document Encoder**.
>
> - The factor **“the LLM’s ability to process external information”** is captured by the **RAG Capability Embedding Layer.**
>
> - The factor **“the query’s role in guiding knowledge retrieval”** is represented by the **Cross Encoder.**
>
> We will incorporate these clarifications into the revised manuscript.
>
> ---
>
> > **Q2**: Include very strong closed-source models.
>
> **A2**: We have followed your suggestion to **add the powerful closed-source LLMs, including GPT-4o and Qwen2.5-Max**, to the original candidate LLM pool with parameter size $\ge$ 32B. We then evaluate our RAGRouter on NQ and WebQ, comparing its performance against non–RAG-aware baselines, including KNN Router and MF.
>
> |           Method            |    NQ     |   WebQ    |    Avg    |
> | :-------------------------: | :-------: | :-------: | :-------: |
> |    Qwen2.5-32B-Instruct     |   44.58   |   50.42   |   47.50   |
> |    Qwen2.5-72B-Instruct     |   40.00   |   48.75   |   44.38   |
> |   Llama-3.3-70B-Instruct    |   51.67   |   50.42   |   51.05   |
> |         Qwen2.5-max         |   51.25   |   52.08   |   51.67   |
> | GPT-4o |   58.33   |   57.08   |   57.71   |
> |         KNN Router          |   57.08   |   50.42   |   53.75   |
> |             MF              |   57.08   |   55.83   |   56.46   |
> |    **RAGRouter (Ours)**     | **58.75** | **57.92** | **58.34** |
>
> Evaluation results show that although **GPT-4o is the top-performing individual model**, **our RAGRouter delivers an additional +0.63% improvement**, while **invoking GPT-4o in only 52.3% of samples**. **RAGRouter further outperforms Qwen2.5-Max (+6.67%), KNN Router (+4.59%), and MF (+1.88%)**. These results highlight RAGRouter’s effectiveness even in the presence of highly powerful LLMs, **leveraging model complementarity to surpass each single LLM, with better latency and cost efficiency by selectively using GPT-4o**.
>
> ---
>
> > **Q3**: Experiments on capability vector dimensionality.
>
> **A3**: We have conducted **hyperparameter studies on varying capability vector dimensionality in Appendix E (Figure 9 and Table 9) of the submission, investigating how different embedding dimensions affect performance across five different datasets.** A summary of the results is shown below:
>
> | Dimension |  128  |  256  |  384  |  512  |  768  | 1024  |
> | :-------: | :---: | :---: | :---: | :---: | :---: | :---: |
> |    Avg. over 5 Datasets    | 63.00 | 63.65 | 63.43 | 63.72 | 64.46 | 63.29 |
>
> As 768-dimensional embeddings yielded the best overall performance, we adopted this setting in the main experiments.
>
> ---
>
> > **Q4**: Will the dataset used to train RAGRouter be open-sourced?
>
> **A4**: We will release the dataset used to train RAGRouter, along with detailed descriptions of its construction and usage instructions.

---

> > ### Comment · Reviewer_s6eC · 2025-08-01
> >
> > Thank you for the response. The additional experiments definitely help improve the completeness of the paper, and I hope the authors will include this part in the final version. Additionally, the authors may still consider using reasoning-based models as future work, since they currently represent the strongest generators. Overall, the response has addressed most of my concerns, and I will maintain my positive score.

---

> > > ### Author Response · Authors · 2025-08-02
> > >
> > > Dear Reviewer s6eC,
> > >
> > > We sincerely appreciate your constructive comments and positive recognition of our work. We’re glad that our response has addressed most of your concerns, and we will include the additional experiments in the final version as suggested.
> > >
> > > To further address your remaining suggestion regarding **adding reasoning-based models, we have added DeepSeek-R1 into the previous evaluation on the powerful closed-source LLMs.** The results are updated as follows:
> > >
> > > |         Method         |    NQ     |   WebQ    |    Avg    |
> > > | :--------------------: | :-------: | :-------: | :-------: |
> > > |  Qwen2.5-32B-Instruct  |   44.58   |   50.42   |   47.50   |
> > > |  Qwen2.5-72B-Instruct  |   40.00   |   48.75   |   44.38   |
> > > | Llama-3.3-70B-Instruct |   51.67   |   50.42   |   51.05   |
> > > |      Qwen2.5-max       |   51.25   |   52.08   |   51.67   |
> > > |         GPT-4o         |   58.33   |   57.08   |   57.71   |
> > > |      DeepSeek-R1       |   56.67   |   53.75   |   55.21   |
> > > |       KNN Router       |   58.75   |   51.67   |   55.21   |
> > > |           MF           |   57.50   |   55.42   |   56.46   |
> > > |  **RAGRouter (Ours)**  | **59.58** | **59.17** | **59.38** |
> > >
> > > We can find although **DeepSeek-R1 is not the top-performing individual model (2.5% lower than GPT-4o)**, its inclusion helps **our RAGRouter outperform GPT-4o by from +0.63% to +1.67%, while reducing GPT-4o usage from 52.3% to 44.79%.** **RAGRouter still significantly outperforms non–RAG-aware baselines such as KNN Router (+4.17%) and MF (+2.92%)**. These results demonstrate that RAGRouter can effectively integrate reasoning-capable models, leveraging their complementarity to improve both performance and efficiency.
> > >
> > > Thank you again for your thoughtful suggestions and continued support!

---

### Official Review · Reviewer_rFRH · 2025-07-02

**Clarity:** 3
**Significance:** 3
**Originality:** 3
**Rating:** 4
**Confidence:** 4

**Summary:**

This paper addresses the challenge of routing user queries—given specific retrieved documents—to the most suitable LLM among multiple candidates. Existing routing techniques focus on static parametric knowledge and fall short in RAG settings, where dynamically retrieved documents can significantly alter model behavior. The authors propose RAGRouter, a routing framework that incorporates both document semantics and model-specific RAG capabilities through a contrastive learning objective. Extensive experiments across five knowledge-intensive benchmarks demonstrate significant performance improvements over both individual LLMs and non-RAG-aware routing baselines.

**Questions:**

I am curious whether the router can maintain high routing accuracy on domain-specific queries that are underrepresented or absent in the training data. Additional evaluations—beyond the medical domain—on domain transferability or robustness to domain shift would help us better understand the generalizability of the proposed contrastive learning method.

**Ethical Concerns:**

["NO or VERY MINOR ethics concerns only"]

**Final Justification:**

The authors’ response has effectively addressed my concerns. The proposed method demonstrates state-of-the-art performance, strong robustness to noisy documents, and better generality under domain shifts. I recommend accepting this paper.

**Quality:**

3

**Strengths And Weaknesses:**

# Strengths
- This paper formalizes the RAG-aware LLM routing problem by taking into account the effect of retrieved documents and the varying RAG capabilities of different LLMs, in contrast to previous works about LLM routing.
- The introduction of learnable RAG capability embeddings is a well-motivated and innovative design that enables RAG-aware routing decisions.
- The proposed method, RAGRouter, significantly outperforms existing baselines across five benchmarks, demonstrating strong performance and good generalization.
- The extended threshold-based mechanism to trade off performance and latency is a practical addition.

# Weakness
- The main contribution and novelty of this paper lies in learning the RAG Capability Embedding to enable RAG-aware routing. However, the paper lacks a dedicated ablation study to validate the effectiveness and necessity of this component. Therefore, the core contribution and motivation of the proposed method remain empirically unverified. （I will consider raising my score if the authors address this concern）
- The paper lacks detail qualitative or quantitative analysis of cases where RAGRouter fails to select the correct LLM or route correctly, especially in comparison with traditional LLM routing methods. For example, the paper does not explicitly evaluate how RAGRouter performs when faced with noisy retrieved documents.

---

> ### Author Rebuttal · Authors · 2025-07-30
>
> Thank you for your thoughtful feedback. We have addressed each of your concerns as follows.
>
> > **Q1**: Ablation study of the RAG Capability Embedding.
>
> **A1**: We have followed your suggestion to add an ablation study to validate the effectiveness and necessity of explicitly modeling the LLM’s intrinsic RAG capability.
>
> In the ablation setting “w/o RAG Capability Embedding”, **we removed the explicit modeling of RAG capabilities** and instead assigned each LLM a single embedding that captures its overall behavior, without distinguishing between parametric knowledge and RAG capabilities.
>
> |                                         | PopQA (Local) | PopQA (Online) | MedMCQA (Local) | MedMCQA (Online) |      NQ       |     WebQ      |      TQA      |      Avg      |
> | :-------------------------------------: | :-----------: | :------------: | :-------------: | :--------------: | :-----------: | :-----------: | :-----------: | :-----------: |
> |                RAGRouter                |     48.52     |     52.59      |      71.48      |      74.44       |     56.67     |     56.67     |     90.83     |     64.46     |
> | w/o RAG Capability Embedding ($\Delta$) | 48.52 (-0.00) | 52.22 (-0.37)  |  71.11 (-0.37)  |  74.44 (-0.00)   | 55.42 (-1.25) | 54.17 (-2.50) | 90.00 (-0.83) | 63.70 (-0.76) |
>
> The ablation study results show that **without the RAG Capability Embedding, the performance across 5 datasets with local and online retrieval strategies drops by 0.76% in average**. Notably, on **NQ and WebQ, which involve imperfect retrieval results, the performance drops by 1.25% and 2.5%**, respectively. These findings validate the effectiveness of our **explicitly introduced and disentangled RAG capability representation**, which plays a critical role in enabling effective RAG-aware routing by accounting for the LLM’s ability to process external information in dynamic knowledge update scenarios.
>
> ---
>
> > **Q2**: Failure and correct case analysis.
>
> **A2**: We have followed your suggestion to add the quantitative analysis of cases where RAGRouter and traditional LLM routing methods fail to select the correct LLM or route correctly on the datasets of PopQA (Local), MedMCQA (Local), and NQ. In particular, we categorize non-trivially unsolvable failures (i.e., excluding cases where all LLMs fail) into 4 types as follows:
>
> - F1: Failure to perceive performance degradation caused by RAG (i.e., cases where LLM's performance decreases after RAG is applied).
>
> - F2: Inherent task difficulty, where the majority of LLMs (e.g., >80%) fail, leading to routing confusion.
>
> - F3: Overconfident selection of stronger LLMs for cases where high-capacity LLMs are chosen but still fail, outside the conditions of F1 and F2.
>
> - F4: Other factors, such as outliers or ambiguous inputs.
>
> We present the type-wise failure rates (as a percentage of all cases) and the overall failure rate (FR) in the table below.
>
> |        Method        |  F1 (%)  |  F2 (%)  |  F3 (%)  |  F4 (%)  |  FR (%)   |
> | :------------------: | :------: | :------: | :------: | :------: | :-------: |
> |      KNN Router      |   5.00   |   8.33   | **1.28** |   1.54   |   16.15   |
> |          MF          |   5.51   |   8.21   |   1.41   |   1.79   |   16.92   |
> | **RAGRouter (Ours)** | **3.85** |   **7.05**   |   1.41   | **0.64** | **12.95** |
>
> We can observe that **compared with traditional LLM routing methods**, including KNN Router and MF, **our RAGRouter achieves** the lowest overall failure rate and **the lowest F1-type failure rate**. This suggests that **RAGRouter more correctly captures performance shifts in LLMs introduced by RAG**. We have also **provided detailed cases in Appendix F where RAGRouter selects correctly while traditional LLM routing methods fail**.
>
> We further examine cases where **RAGRouter fails while traditional LLM routing methods succeed**. We find that **such cases are less than 2% of all the evaluated cases**. Notably, **more than 50% of these cases fall into the F2 type**, where **the performance gap between candidate LLMs is inherently small**.
>
> ---
>
> > **Q3**: Performance when faced with noisy retrieved documents.
>
> **A3**: **We have reported and analyzed RAGRouter’s performance under noisy retrieved documents in Appendix C (Routing Performance under Noisy Retrieval) of the submission.** In particular, we add synthetic noise into the NQ, WebQ, and TriviaQA datasets. Take TriviaQA for example. The evaluation results under four conditions, including Golden Context, Relevant Noise, Irrelevant Noise, and Counterfactual Noise, are summarized as follows.
>
> |        Method        | Golden Context | Relevant Noise | Irrelevant Noise | Counterfactual Noise |    Avg    |
> | :------------------: | :------------: | :------------: | :--------------: | :------------------: | :-------: |
> |      KNN Router      |     96.67      |     78.33      |      83.33       |        86.67         |   86.25   |
> |          MF          |     95.00      |     76.67      |      80.00       |        80.00         |   82.92   |
> | **RAGRouter (Ours)** |   **98.33**    |   **86.67**    |    **90.00**     |      **88.33**       | **90.83** |
>
> We can observe that **our RAGRouter consistently outperforms traditional LLM routing methods, with performance gains more than 4.58%, highlighting its strong robustness to retrieval noise**.
>
> ---
>
> > **Q4**: Additional evaluations on domain transferability or robustness to domain shift.
>
> **A4**: To evaluate RAGRouter’s generalization across domains, we design two cross-domain settings:
>
> - Setting 1: **Trained on MedMCQA (Local) that falls into medical domain, but tested on a new dataset HotpotQA for Wikipedia-based multi-hop QA**.
>
> - Setting 2: **Trained on NQ that contains real-world search queries, but tested on MedMCQA (Local)**.
>
> |            | Setting 1: MedMCQA (Local) $\to$ HotpotQA | Setting2: NQ $\to$ MedMCQA (Local) |
> | :------------------: | :--------------------------------------: | :--------------------------------: |
> |      KNN Router      |                  18.60                   |               63.33                |
> |          MF          |                  20.20                   |               58.89                |
> | **RAGRouter (Ours)** |                **21.20**                 |             **68.89**              |
>
> Evaluation results show that **RAGRouter still outperforms traditional LLM routing methods by 1.00% and 5.56% in the two cross-domain settings, demonstrating strong domain generalization.** This performance advantage can be attributed to **RAGRouter’s contrastive learning framework, which effectively captures relative RAG capability differences among candidate LLMs, even when transferred across domains.**

---

> > ### Comment · Reviewer_rFRH · 2025-08-06
> >
> > Thank you for conducting the additional experiments. The results have addressed my concerns effectively. I will raise the Quality score to 3, and increase my rating and confidence to 4.

---

> > > ### Author Response · Authors · 2025-08-07
> > >
> > > Thank you very much for your kind and encouraging feedback. We're glad to hear that the additional experiments have addressed your concerns. We sincerely appreciate your thoughtful review and the improved scores, and will carefully incorporate the suggested revisions into the final version.

---

### Note · Authors · 2025-08-12

Dear Reviewers, ACs, and SACs,

We sincerely thank you for the constructive and insightful feedback. We are glad that our response has addressed all the concerns raised. We greatly appreciate all the positive evaluations and strong support. In particular, we are truly grateful for the reviewers’ acknowledgment of the following highlights and key contributions of our work:

+ We, for the first time, study the new RAG-aware LLM routing problem by modeling capability shifts in LLMs induced by external retrieved documents. (`Reviewers rFRH, WwUD`)

+ We propose a new router network architecture RAGRouter, which introduces document encoder and RAG capability embedding layer to model how LLMs absorb and respond to retrieved evidence. Further considering the retrieval setting can substantially alter an LLM’s answerability by shifting the knowledge representation, we propose intra- and cross-setting contrastive learning objectives to capture the performance variations with and without RAG settings. (`Reviewers rFRH, SuEE, WwUD`)

+ We extensively evaluate RAGRouter across open-source, closed-source, and reasoning-based LLMs (`Reviewers s6eC, SuEE`) on diverse tasks (`Reviewer SuEE`), showing significant improvements over traditional LLM routing baselines and each individual LLM. Ablation studies, including isolating the RAG capability embedding, validate the necessity of each design component (`Reviewers rFRH, SuEE`). Cross-domain evaluation (`Reviewer rFRH`), performance under noisy retrieved documents (`Reviewers rFRH, WwUD`), as well as failure and correct case analysis (`Reviewer rFRH`) demonstrate the generalization and the robustness of our RAGRouter.

Once again, we sincerely appreciate your generous time, dedicated effort, and constructive feedback. Your insights have been invaluable in refining and strengthening our work, and we are truly grateful for your thoughtful engagement throughout the review process.

Best regards,

Authors of Paper #11817

---

### Decision · Program_Chairs · 2025-09-17

**Decision:**

Accept (poster)

**Comment:**

This paper introduces RAGRouter, an approach for routing queries to the most suitable LLM in a RAG setting.

The reviewers noted the following strengths of the paper:
* The proposed approach is novel. It is the first to define and address the problem of LLM routing for RAG
* The proposed architecture is technically sound and well-motivated. It features an innovative learnable RAG capability embedding for each target LLM and a unique contrastive learning objective aimed at capturing shifts in knowledge
* The experimental evaluation is extensive and clearly demonstrates the strong effectiveness of the proposed approach across a diverse range of knowledge-intensive tasks
* The framework has a practical score threshold mechanism that makes it possible to tradeoff between quality and latency

The reviewers also identified a number of weakness with the paper, including:
* The overall innovation could be viewed as incremental, as it builds on top of existing contrastive learning and model routing approaches
* The experiments reported in the initial version of the paper were limited in scope, lacking evals using SOTA models like GPT-4o and did not include non-QA tasks

The authors submitted an exceptionally thorough rebuttal. Following the rebuttal, the most engaged reviewers confirmed their favorable view of the paper, which moved the paper from a borderline case to a clear consensus.

Overall, this is a high-quality paper that puts forth a strong solution to a novel and highly timely problem. It should therefore be accepted.